# Drought years in peatland rewetting: Rapid vegetation succession can maintain the net $CO_2$ sink function

Florian Beyer[1,*], Florian Jansen[2], Gerald Jurasinski[2], Marian Koch[3,†], Birgit Schröder[2], and Franziska Koebsch[2,*]

[1]Geodesy and Geoinformatics, Faculty for Agricultural and Environmental Sciences, Rostock University, 18059 Rostock, Germany
[2]Landscape Ecology, Faculty for Agricultural and Environmental Sciences, Rostock University, 18059 Rostock, Germany
[3]Soil Physics, Faculty for Agricultural and Environmental Sciences, Rostock University, 18059 Rostock, Germany
[†]deceased, 15 April 2020
[*]These authors contributed equally to this work.

**Correspondence:** florian.beyer@uni-rostock.de

**Abstract.** The rewetting of peatlands is regarded as important nature-based climate solution and intended to reconcile climate protection with the restoration of self-regulating ecosystems that are resistant to climate impacts. Although the severity and frequency of droughts is predicted to increase as a consequence of climate change, it is not well understood whether such extreme events can jeopardize rewetting measures. The goal of this study was to better understand drought effects on vegetation development and the exchange of the two important greenhouse gases $CO_2$ and $CH_4$ especially in rewetted fens. Based on long-term reference records, we investigated anomalies in vegetation dynamics, $CH_4$ emissions, and net $CO_2$ exchange, including the component fluxes ecosystem respiration ($R_{eco}$) and gross ecosystem productivity (GEP), in a rewetted fen during the extreme European summer drought 2018. Drought-induced vegetation dynamics were derived from remotely sensed data.

Since flooding in 2010, the fen was characterized by a patchy mosaic of open water surfaces and vegetated areas. After years of stagnant vegetation development, drought acted as a trigger event for pioneer species such as *Tephroseris palustris* and *Ranunculus sceleratus* to rapidly close persistent vegetation gaps. The massive spread of vegetation assimilated substantial amounts of $CO_2$. In 2018, the annual GEP budget increased by 20 % in comparison to average years (2010–2017). $R_{eco}$ increased even by 40 %, but enhanced photosynthetic $CO_2$ sequestration could compensate for half of the drought-induced increase in respiratory $CO_2$ release. Altogether, the restored fen remained a net $CO_2$ sink in the year of drought, though net $CO_2$ sequestration was lower than in other years. $CH_4$ emissions were 20 % below average on an annual basis, though stronger reduction effects occurred from August onwards, when daily fluxes were 60 % lower than in reference years.

Our study reveals an important regulatory mechanism of restored fens to maintain their net $CO_2$ sink function even in extremely dry years. It appears that, in times of more frequent climate extremes, fen restoration can create ecosystems resilient to drought. However, in order to comprehensively assess the mitigation prospects of peatland rewetting as nature-based climate solution, further research needs to focus on the long-term effects of such extreme events beyond the actual drought period.

# 1 Introduction

Peatlands constitute the largest terrestrial C store and exert significant feedback effects on the climate system (Gorham, 1991; Frolking and Roulet, 2007; Yu et al., 2010). Under the massive human disturbance of recent times, the global peatland biome has shifted from a net sink to a source of greenhouse gases (GHG) (Leifeld et al., 2019). The shift in peatlands climate

function is mainly a result of extensive drainage: when water levels fall, oxygen availability initiates a cascade of organic matter breakdown that culminates in peat decomposition (Freeman et al., 2004; Fenner and Freeman, 2011). In this way, drainage turns peatlands from $CO_2$ sinks to $CO_2$ sources. Among minerotrophic peatlands (fens) in mid Europe, 90 % have been drained, most of them for agricultural purposes (Pfadenhauer and Grootjans, 1999; Moen et al., 2017). Drained peatlands rank among the largest $CO_2$ sources from agriculture and forestry in many European countries, even when they cover only a

small percentage of the national areas (Tiemeyer et al., 2016; Tubiello et al., 2016). A reduction of these emissions is urgently required because drained peatlands consume 10–41 % of the remaining emission budget to maintain global warming below 2° C (Leifeld et al., 2019).

Rewetting is a common measure, not only to restore the natural habitat function of peatlands, but also to stop $CO_2$ emissions and thereby to mitigate climate change (Leifeld and Menichetti, 2018). Peatland conservation and rewetting is therefore

considered one of the major natural climate solutions (Griscom et al., 2017; Leifeld and Menichetti, 2018) and a key measure to turn the terrestrial land system to its natural net $CO_2$ sink function (Humpenöder et al., 2020). As rewetting re-establishes anaerobic conditions, it diminishes $CO_2$ emissions from peat degradation. However, rewetting may also resume the emissions of methane ($CH_4$), a strong, yet short-lived greenhouse gas (Wilson et al., 2016). The net cooling effect of peatland rewetting is essentially accomplished by the savings of $CO_2$ emissions, which is why climate mitigation measures in peatlands focus

primarily on the reduction of the $CO_2$ source (Tiemeyer et al., 2020). However, the warming pulse caused by concurrent $CH_4$ emissions can retard the desired mitigation effect (Günther et al., 2020).

The successful implementation of peatland rewetting can be challenging, as the degradation processes provoked by drainage are largely irreversible. Under intense compaction and decomposition, the peat surface can subside for several decimeters (Leifeld et al., 2011) and rewetted fen areas can easily develop to shallow lakes with average water depths of 20—60 cm

(Steffenhagen et al., 2012). Slow or stagnant vegetation development withholds the extensive spread of peatland species as prerequisite for $CO_2$ uptake and C accumulation (Timmermann et al., 2009; Koch et al., 2017).

Given the importance of hydrological conditions for peat conservation and formation, also meteorological drought can severely impact peatland functioning (Dise, 2009). In analogy to human-induced drainage, drought implies a lowering of the ground water level which may enhance ecosystem respiration ($R_{eco}$) and peat consumption (Alm et al., 1999; Knorr et al.,

2008; Lund et al., 2012). Further, gross ecosystem productivity (GEP) may decrease as plant stress due to drought limits photosynthetic $CO_2$ uptake (Shurpali et al., 1995; Schreader et al., 1998; Arneth et al., 2002; Lafleur et al., 2003; Lund et al., 2012; Olefeldt et al., 2017). At the same time, temporary drought can lower the obligate anaerobic production of $CH_4$ (Morozova and Wagner, 2007; Knorr et al., 2008) and increase the oxic consumption of $CH_4$ in the peat areas fallen dry (Ma et al., 2013). Altogether, years of drought may reduce $CH_4$ emissions and turn peatlands from net $CO_2$ sinks to sources of

$CO_2$ (Lafleur et al., 2003; Lund et al., 2012), whereby the magnitude of effects can be further modulated by plant community composition (Robroek et al., 2017).

Worldwide 43–51 Mha of peatlands are drained (Joosten et al., 2016; Leifeld and Menichetti, 2018; Leifeld et al., 2019). Rewetting these areas is essential to achieve our climate goals (Humpenöder et al., 2020; Günther et al., 2020). However, estimates on the mitigation potential of nature based climate solutions often lack any consideration for how future climate

change will impact peatland functioning and greenhouse gas exchange. In view of increasing frequency and severity of climatic extreme events (Pachauri et al., 2014), drought has the potential to jeopardize the climate mitigation goals of peatland rewetting (Lavendel, 2003; Harris et al., 2006). Yet, our understanding of drought effects on rewetted peatlands is largely incomplete, which adds considerable uncertainty on the mitigation potential achievable through natural climate solutions under a changing climate. The majority of drought studies are designed as mesocosm and/or treatment experiments and address near-natural

bogs (Shurpali et al., 1995; Alm et al., 1999; Arneth et al., 2002; Lafleur et al., 2003; Lund et al., 2012). As hydrological and vegetation differ between peatland types, the same drought-related mechanisms may not necessarily occur in fens (Sulman et al., 2010). Even comparisons with pristine fens may be misleading, because the drainage-rewetting sequence irreversibly affects ecosystem functioning of restored fens (Koch et al., 2017). Hence, a better understanding of drought-induced processes in restored fens is needed.

Here, we aim to elucidate the in situ effects of drought on vegetation development, as well as the exchange of $CO_2$ and $CH_4$ in rewetted fens. To this end, we investigated the impact of the extreme summer drought 2018 on a rewetted degraded fen in north eastern Germany. The drought event caused the water level to drop below the ground surface, for the first time since rewetting and therefore provided a good opportunity to investigate our research question. Vegetation development and the exchange of $CO_2$ and $CH_4$ in our particular fen site have been monitored since the rewetting started in 2010, which offers a

valuable long-term reference record for the assessment of drought-induced effects. Vegetation dynamics were evaluated both, on canopy and species level. For the canopy level we used satellite-derived remote sensing products such as the enhanced vegetation index (EVI) and the fraction of absorbed photosynthetically active radiation (fPAR). Information on species level were obtained through vegetation mapping derived from multi-sensor data of an unmanned aerial system (UAS). Drought effects on greenhouse gas exchange, including the $CO_2$ component fluxes $R_{eco}$, and GEP were investigated based on a multi-

80 year record of eddy covariance measurements (Montgomery, 1948; Baldocchi, 2003). The $CO_2$ flux time series was also used to infer the start and end of the carbon uptake period (CUP) as proxy to derive drought effects on plant phenology. In addition, we deployed a simple GEP light-use efficiency model (Hunt JR, 1994; Gower et al., 1999) to further elucidate the biophysical mechanisms that control photosynthetic $CO_2$ uptake during periods of drought. This interdisciplinary long-term approach, including ecosystem-scale monitoring of vegetation development and greenhouse gas exchange, allowed us to track

the response mechanisms of a rewetted fen to a severe drought event and thereby to infer insights about the resilience of this novel ecosystem in times of more frequently upcoming climate extremes.

## 2  Methods

### 2.1  Site description

The study area "Rodewiese" (WGS84: N 54.211°, E 12.178°) is a coastal paludification fen in the nature reserve "Heiligensee und Hütelmoor", located in north eastern Germany (Figure 1). The area has been heavily drained for grassland use since the

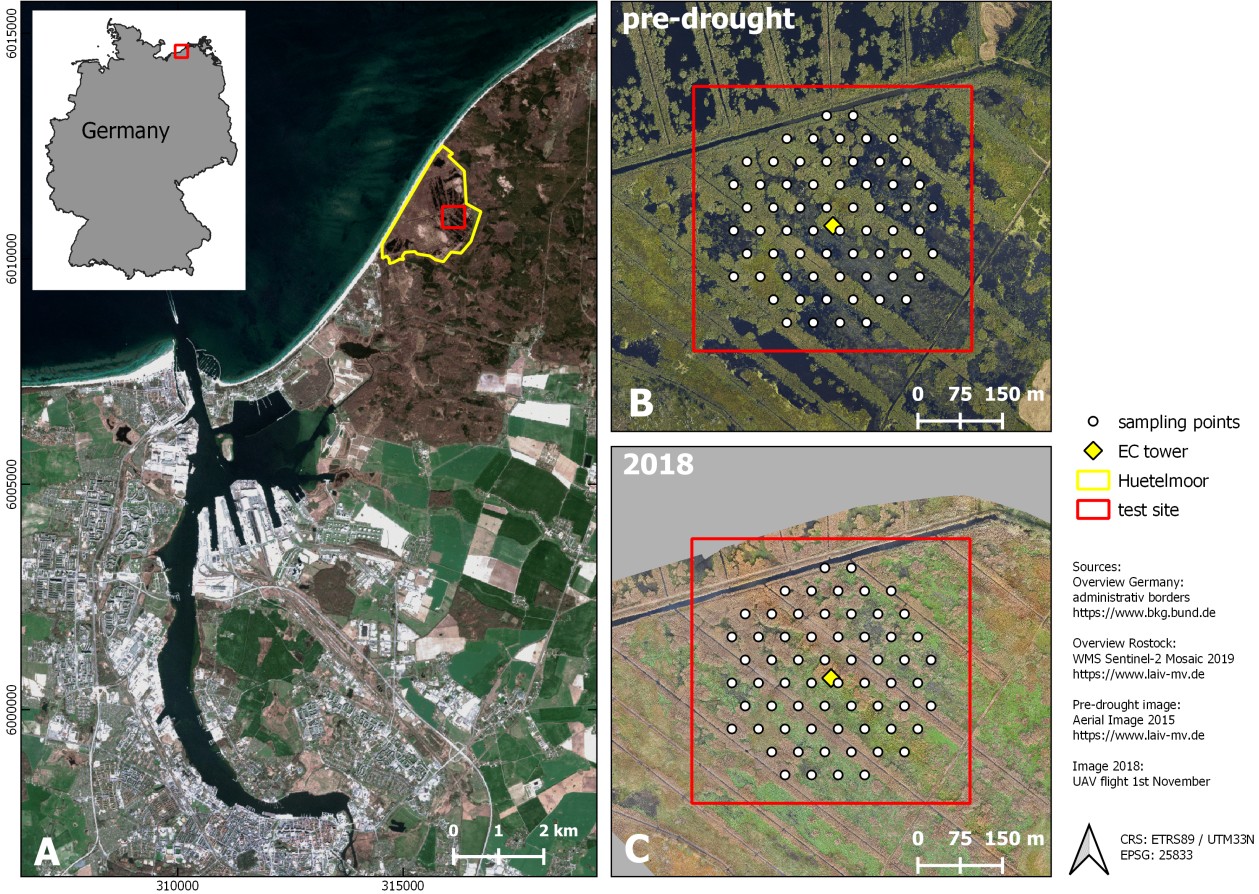

**Figure 1.** Study site. A: Location (City of Rostock). B (August 2015) and C (November 2018): Aerial photograph with vegetation survey grid. From 2010 to 2017 (pre-drought), the fen was almost permanently inundated. At that time, the canopy consisted of a patchy mosaic of open water and vegetated areas. During the drought 2018, the site fell completely dry, except for the former drainage ditches.

1970s with water levels down to 1.6 m below ground. Under drainage, the peat was degraded strongly, and can, nowadays, be described as sapric histosol. In winter 2009/2010, the site was rewetted with the goal to stop peat decomposition and to create a self-regulating ecosystem and water fowl habitat. As a result of rewetting, the site became inundated year-round and the canopy turned to a patchy mosaic of different dominant species and open water areas. Since then, the vegetation was dominated by

stands of competitive emergent macrophytes such as Common Reed (*Phragmites australis*) and Lesser-Pond sedge (*Carex acutiformis*) as well as Grey and Sea Club rush (*Schoenoplectus tabernaemontani* and *Bolboschoenus maritimus*). Both of the two latter species present relics of former brackish impact from the near-by Baltic sea. Vegetation patterns were mostly stable in the years following inundation with a slight tendency towards higher patch compactness. Koch et al. (2017) provide a detailed description of the vegetation development of 2011 until 2014.

## 2.2 Assessing canopy dynamics

Satellite-derived vegetation indices provide information on plant phenology and coverage on canopy level, the spatial scope of which fits well to that of the eddy covariance approach. For this study, we obtained the enhanced vegetation index (EVI) and the fraction of absorbed photosynthetically active radiation (fPAR) from MODIS (Moderate Resolution Image Spectrometer). The EVI is especially suited to resolve variations at the upper end of the canopy reflection range (Huete et al., 2002) and has been successfully used in past studies to describe subtle vegetation dynamics in our study area (Koebsch et al., 2013).

EVI data were retrieved from the MOD13A1 and MYD13A1 product, and fPAR data were retrieved from MCD15A3H, using the NASA AppEEARS tool, respectively (lpdaacsvc.cr.usgs.gov/appeears/). The time series created spanned the period 2010–2018 and the 500 m pixel size covered the eddy covariance flux climatology ( Figure 1 and Figure B2, Appendix B2). We combined data from both MODIS satellites, Aqua and Terra, and thereby obtained time series with 8 day intervals for EVI and 4 day intervals for fPAR. The data records were filtered according to pixel reliability and pixel-wise quality assessment. Subsequently, data gaps were filled by linear interpolation and the time series was smoothed with an exponentially-weighted function (span = 5) to reduce unwanted scatter.

## 2.3 Vegetation mapping

### 2.3.1 Preprocessing of the unmanned aerial system data

Unmanned aerial system data were collected to classify plant composition and distribution of the dominant species. In order to assess the drought effect on vegetation, the changes observed in 2018 were related to the state prior to drought as described in Koch et al. (2017). Accordingly, the study area and processing routines for 2018 were harmonized to the best possible degree with the approach used in Koch et al. (2017). In contrast to Koch et al. (2017) not only normal RGB data and texture indices were available but also additional sensors as well as data types (additional wavelengths and geometrical information) were used.

Aerial images were acquired in late autumn (1 November 2018) using an fixed-wing unmanned aerial system (UAS, Sensefly eBee Plus). As the UAS can operate only one camera at a time, high-resolution true color images (SenseFly S.O.D.A, 20 Mpix), multispectral images (Parrot Sequoia, 4x 1.2 Mpix) and thermal images (SenseFly ThermoMap, 0.3 Mpix) were taken during subsequent flights within a time frame where insolation can be considered as stable. The acquired images were then mosaiced with the photogrammetric software Pix4D (Figure B1, appendix B1). The multisensor data set was processed as described in Beyer et al. (2019) and, eventually, consisted of 107 bands: 3 RGB bands, 4 multispectral bands, and 1 thermal band, as well

as 1 digital surface model (DSM), 74 spectral and 24 textural indices. The DSM was derived photogrammetrically using RGB color information (Figure B1) and can, due to the flat topography of the study area, be interpreted as plant height proxy. The texture indices were calculated as in Koch et al. (2017) for each RGB band. The 74 spectral indices were selected using the Index Database (www.indexdatabase.de, Henrich et al. (2012, 2009)). The main reason to select such a high number of spectral indices was not only to improve the classification accuracy but especially to get better knowledge of the importance of the specific wavelengths used within the multisensor data set. This approach continues the earlier study from Beyer et al. (2019). All bands, indices and their meaning are listed in Appendix B3 (Table B1). Further, a Python script and an overview of the used indices can be found on github.com/florianbeyer/SpectralIndices.

### 2.3.2 Vegetation survey

Likewise, with the study of Koch et al. (2017), vegetation sampling in 2018 was conducted within an equidistant grid of 64 circular plots, each with a 1m radius (Figure 1). The re-survey was conducted at the end of September and included total plant coverage as well as species coverage (%). Among the 36 species found, only *Phragmites australis*, *Schoenoplectus tabernae-montani*, *Bolboschoenus maritimus*, *Tephroseris palustris*, *Ranunculus sceleratus*, and *Carex acutiformis* were occurring in dominant stands. Here, dominance was defined by (1) the per-plot-abundance and (2) the occurrence frequency across all 64 sample points (more than 30 times occurred in 65 plots or more than 50 % occurrence per plot). These six dominant species were, in concert with bare peat and open water, incorporated as surface classes in the following analysis.

### 2.3.3 Vegetation classification

To classify the vegetation cover, we used the Random Forest (RF, Breiman (2001)) classifier with 500 trees and a minimum branching depth of 2. RF has proven to be a robust and efficient machine learning classification approach in previous remote sensing studies (Beyer et al., 2015; Belgiu and Drăguţ, 2016; Beyer et al., 2019). On the basis of the vegetation mapping, a calibration data set was generated in GIS in order to train the RF. We assessed the performance of the RF model with an independent validation data set. The RF classification algorithm achieved an overall accuracy of 99.84 %. Also, the single class accuracies were high and ranged between 98 and 100 %. In addition, we extracted the importance of every single band in the multisensor data set using the GINI coefficient (Archer and Kimes, 2008) in order to assess the most important input variables. The results of the importance analysis is summarized in Table B2 (Appendix B3). The classification script can be found at github.com/florianbeyer/RandomForest-Classification.

### 2.4 $CO_2$ flux processing

The exchange of $CO_2$ and $CH_4$ was determined with the eddy covariance approach, which provides a continuous time series of half hourly fluxes on ecosystem scale. The setup comprised open-path sensors for $CO_2$ and $CH_4$ molar density (LI-7500) and LI-7700 from LI-COR, Lincoln, NE, USA), and a three-dimensional sonic anemometer (CSAT3, Campbell Scientific, Logan, UT, USA) measuring wind velocities and sonic temperature. All signals were recorded by a CR3000 Micrologger (Campbell

Scientific, Logan, Utah) with a scan rate of 10 Hz. Half-hourly fluxes of $CO_2$ and $CH_4$ were processed with the software EddyPro version 6.0.0 (LI-COR, Lincoln, NE, USA) using the common corrections for open path eddy covariance set ups.

Refer to Koebsch et al. (2013) and Koebsch et al. (2015) for more details on the setup and the complete sequence of flux processing steps. The source area of the measured greenhouse gas fluxes was determined with the analytical footprint model of Kormann and Meixner (2001) and cumulated over the course of the year. According to the resulting footprint climatology, 90 % of the measured gas exchange comes from within 200 m distance around the eddy covariance tower (Figure B2, Appendix B2).

Data gaps in the $CO_2$ and $CH_4$ flux time series were filled using artificial neural networks (ANNs, Bishop (1995)) based on the common back propagation algorithm incorporated in the R package neuralnet (R Core Team 2019; Fritsch 2016). Gap filling was conducted in two steps: (1) For small data gaps < 24 hours, we set up several ANNs that predicted half-hourly fluxes separately for each year. (2) For larger data gaps > 24 hours, we aggregated the data set day-wise and set up a single ANN that encompassed all available measurements from 2009 to 2018. Input variables for all ANNs included air temperature,

global radiation, water level, and EVI, as well as fuzzy-transformed variables for time of day and season. A simple architecture comprising one hidden layer and 3–4 nodes proved applicable for all ANNs. Validation of the ANNs with an independent data subset yielded determination coefficients ranging from 0.46–0.83 for half hourly fluxes and 0.77–0.93 for daily aggregated fluxes.

The net ecosystem exchange of $CO_2$ (NEE) was further partitioned into its two component fluxes gross ecosystem produc-
175 tivity (GEP) and ecosystem respiration ($R_{eco}$, eq. 1).

$$NEE = R_{eco} - GEP \tag{1}$$

Hereby, GEP represents the photosynthetic sequestration of $CO_2$ from the atmosphere into the canopy, whilst $R_{eco}$ represents the $CO_2$ release by autotrophic and heterotrophic respiration into the atmosphere. We partitioned NEE into its component fluxes with an ANN algorithm that predicted $R_{eco}$ from the daily aggregated nighttime fluxes (global radiation threshold < 5 W/m$^2$).

Subsequently, we calculated GEP from the difference between the measured daytime NEE and modeled $R_{eco}$. Input variables for the ANN included air temperature, water level, EVI, as well as fuzzy-transformed variables for different seasons. The ANN was build from one hidden layer and 4 nodes. Validation of the ANN yielded a determination coefficient for the nighttime fluxes of 0.88.

## 2.5 Auxiliary data

Meteorological measurements since 2009 were conducted directly at the eddy covariance tower and logged in 30 minute intervals. Measurements included (1) global radiation (Rg), measured with a pyranometer (CMP 3; Kipp & Zonen, Delft, the Netherlands), (2) air temperature (HMP45C, Vaisala, Vantaa, Finland) (3) and precipitation (52203 RM Young). Minor Data gaps were filled with data from a nearby station of the German Weather Service (DWD) in 7.5 km distance to our field station (cdc.dwd.de/portal/ Stations-ID: 4271). DWD weather data were also used for the meteorological long-time reference period

1999–2017.

The water level time series was reconstructed back to 2010 from manual discrete measurements and pressure-compensated automated measurements (Onset U20-001-01 Water Level Data Logger, Onset, Bourne, USA). The final water level time series is referenced to the average elevation height of the fen with positive values indicating water levels above surface.

In addition, we used the carbon uptake period (CUP) as proxy to describe potential drought effects on plant phenology. The start and end dates of the CUP were extracted from a 20 day moving window sliding over the time series of daily NEE sums. CUP started from the day on, when the fen acted as a net $CO_2$ sink for at least 20 days in a row, i.e, all daily NEE sums within the moving windows were negative. CUP ended from the day on, when the fen acted as a net $CO_2$ source for at least 20 days in a row, i.e, all daily NEE sums within the moving windows were positive.

## 2.6 Light use efficiency modeling

The light use efficiency (LUE) of GEP relates plant $CO_2$ assimilation to the light absorption capacity of the canopy and has been originally conceived as ecosystem-specific constant (Monteith, 1972; Heinsch et al., 2003). However, LUE also varies over the course of the season and can be attenuated through the plant-physiological response to environmental stresses (Heinsch et al., 2003; Connolly et al., 2009). LUE is given as:

$$GEP = \epsilon * APAR \tag{2}$$

where $\epsilon$ is the light use efficiency parameter ($g\ C\ MJ^{-1}$). GEP is derived from the eddy covariance approach and here implemented in $g\ CO_2\text{-}C\ m^{-2}\ d^{-1}$. APAR is the absolute value of absorbed photosynthetically active radiation (PAR) in $MJ\ m^{-2}\ d^{-1}$ and is given as:

$$APAR = \downarrow PAR * fPAR \tag{3}$$

where ↓PAR is incident photosynthetically active radiation in $MJ\ m^{-2}\ d^{-1}$. FPAR is the remote-sensing derived fraction of the photosynthetically active radiation absorbed by the canopy within the eddy covariance footprint.

## 3 Results and Discussion

### 3.1 Meteorological and hydrological conditions in 2018

At the study site, 2018 was among the warmest and sunniest years within the reference period (1999–2018; Figure 2) with only 2003 sharing the same low precipitation sums (457 mm). Hence, 2018 was also the driest year since rewetting of the fen started in 2010. Mean annual temperature amounted to 10.8 °C which was 1 K above the long term average of the reference period and global radiation in 2018 summed up to 2,370 kW m$^{-2}$ which exceeded the long term radiation sum by 213 kW m$^{-2}$. Total precipitation sum in 2018 was 160 mm below the long term average total of 617 mm (Figure 2B).

Drought, excessive heat and radiation in 2018 occurred primarily from April to July. During these months, the mean temperature exceeded the long term average April–July temperature (14.0°C) by 1.9 K. The global radiation sum during April–July

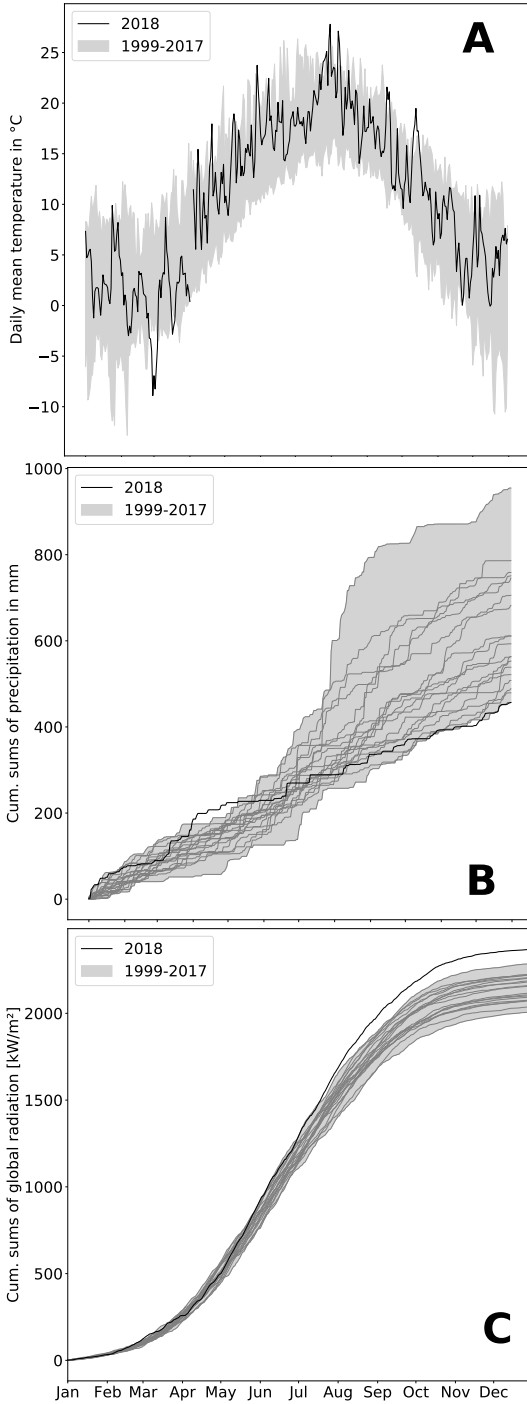

**Figure 2.** Air temperature (A), cumulative precipitation (B) and cumulative global radiation (C) over the course of the year. Variables are represented as black line for 2018 whereas the grey shading represents the variable range (minimum-maximum) throughout the reference period 1999–2017.

**Table 1.** Annual means and sums of certain climatic and other parameters used in the manuscript from 2010–2018 (EVI = enhanced vegetation index, fPAR = fraction of absorbed photosynthetically active radiation, LUE = light use efficiency, CUP = carbon uptake period, doy = doy of year).

| Year | Temperature annual mean (°C) | Precipitation annual sum (mm) | Radiation annual sum (kW/m$^2$) | Water level annual mean (cm) | EVI annual mean | fPAR annual mean | LUE annual mean $g\ C\ MJ^{-1}$ | CUP start DOY | end DOY |
|------|------|------|------|------|------|------|------|------|------|
| 2010 | 8.1 | 706 | 2096.399 | 36 | 0.28 | 0.536 | 0.177 | 145 | 296 |
| 2011 | 9.8 | 955 | 2109.110 | 41 | 0.25 | 0.509 | 0.120 | 113 | 294 |
| 2012 | 9.2 | 490 | 2103.767 | 20 | 0.26 | 0.505 | 0.137 | 136 | 291 |
| 2013 | 9.4 | 611 | 2183.956 | 24 | 0.27 | 0.537 | 0.121 | 142 | 280 |
| 2014 | 10.7 | 553 | 2224.981 | 19 | 0.28 | 0.547 | 0.115 | 114 | 266 |
| 2015 | 10.3 | 611 | 2223.394 | 26 | 0.27 | 0.518 | 0.132 | 130 | 278 |
| 2016 | 10.1 | 479 | 2160.338 | 25 | 0.27 | 0.524 | 0.125 | 131 | 245 |
| 2017 | 10.1 | 746 | 2075.759 | 39 | 0.27 | 0.521 | 0.101 | 138 | 286 |
| 2018 | 10.7 | 457 | 2369.617 | 17 | 0.32 | 0.603 | 0.120 | 130 | 307 |

2018 exceeded the average radiation sum by 140 kW m$^{-2}$ (long term average: 1,277 kW m$^{-2}$). Furthermore, precipitation from April to July 2018 summed up to only 111 mm, which is less than half of the rainfall occurring in average years (228 mm). In particular, May 2018 was extraordinarily dry with only 5 mm of rainfall (average May rainfall: 51 mm).

The spatially averaged, mean annual water level (Figure 3A and Table 1) in 2018 was 17 cm above surface level (a.s.l.) which is in the lower range of post-rewetting water levels (20–40 cm a.s.l. from 2010–2018). However, meteorological conditions induced a pronounced hydrological variation during the course of 2018. As a result of unusually high precipitation in the previous year (746 mm), water level was still extraordinarily high (0.4 m a.s.l.) until early spring 2018 but decreased rapidly due to rainfall deficit starting in April. So there might be the possibility that the filled water reservoirs from 2017's high rates of rainfall contributed to the postponement of the hydrological drought and could thereby buffer the effect of the meteorological drought, at least until April 2018. Whilst the fen had been permanently inundated since the rewetting in 2010, the water level dropped below ground surface in August 2018. A water level minimum of 0.4 m below surface level (b.s.l.) was met in October.

## 3.2 Vegetation response to drought

### 3.2.1 Species shift

Rewetting of the fen in 2010 initiated a shift towards flooding resistant species Koch et al. (2017). However, these dynamics were confined only to the first 1–2 years after rewetting, whilst vegetation development stagnated in the following and provided a stable baseline for the investigation of drought effects. In 2014 (Figure 4a), which serves as reference year for the vegetation situation prior to drought, the fen canopy consisted of *Phragmites australis* (47.8 %), *Schoenoplectus tabernaemontani* (21.0

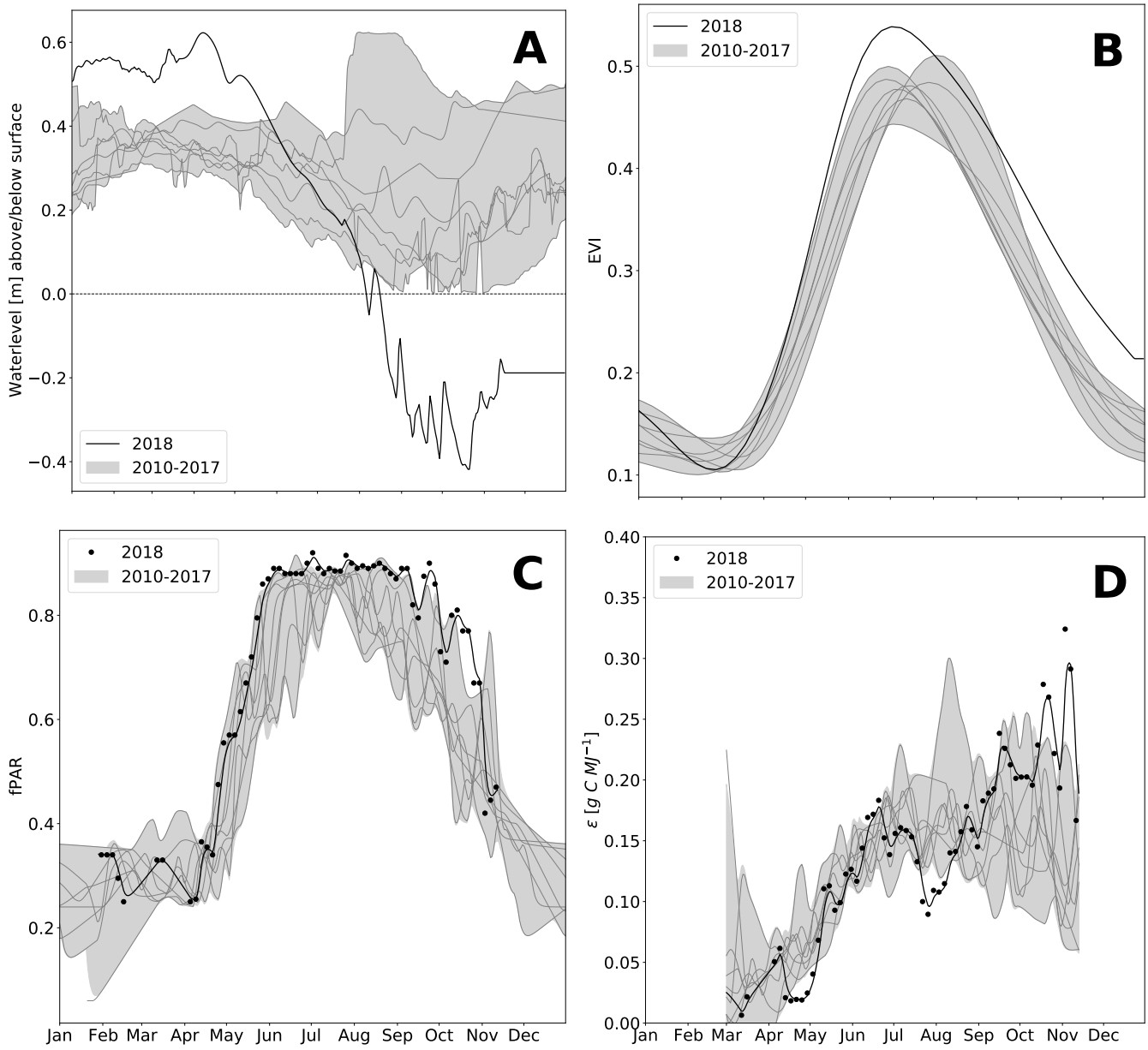

**Figure 3.** Water level (A), enhanced vegetation index (EVI, B), absorbed photosynthetically active radiation (fPAR, C) and light use efficiency ($\epsilon$, D) over the course of the year. Variables are represented as black line for 2018 whereas the grey shading represents the variable range (minimum-maximum) throughout the reference period 2010–2017. Due to large uncertainties occurring during dormancy period, $\epsilon$ is presented only for the growing season.

%), open water (20.5 %) *Carex acutiformis* (5.6 %), and *Bolboschoenus maritimus* (5.1 %). Field observations proved these area proportions to remain stable until 2017. With the exception of Phragmites, which constituted the dominant species (areal proportions of 44.4 %), the drought 2018 dramatically changed the species composition of the site (Figure 4b). When rain failed to fall, open water patches dried up completely and were colonized by *Tephroseris palustris* and *Ranunculus sclera-tus*. Both are pioneer species that can rapidly spread along the nutrient-rich shores of dried-up water bodies (Henker et al., 2006). Though of minor abundance in previous years, (Leipe and Leipe, 2017), in 2018, *Tephroseris palustris* and *Ranunculus sceleratus* gained a spatial coverage of 26.6% within a few weeks. The spatial proportion of both *Bolboschoenus maritimus* and *Schoenoplectus tabernaemontani* decreased from 26.1 down to 6.3 % in 2018. In contrast to previous years, when each of these species formed extensive clusters, they now appeared strongly dispersed and were therefore merged into a single veg-etation class. In contrast, the areal coverage of *Carex acutiformis*, a species adapted to moist conditions, increased from 5.6 to 17.3 %. Hence, after years of stagnation, drought changed the species composition of the fen within weeks: Dried-up open water patches served as habitat for fast-growing pioneer plants, but also the established vegetation responded with substantial withdrawal of flooding-adapted species and a spread of species adapted to moderate moisture.

### 3.2.2  Seasonal dynamics

The special vegetation dynamics during the drought year 2018 were best represented by the enhanced vegetation index (EVI). The EVI increased rapidly from a comparatively low initial value of  0.1 in February/March to a new maximum of 0.53 at the start of July. The steep spring-time rise and the high summer peak in EVI can most likely be attributed to the rapid growth of the established vegetation which was triggered by high temperatures and radiation supply from April on. However, in comparison to other years, EVI decreased early at the beginning of July 2018, which marked the onset of drought-related changes in canopy reflectance when water level dropped below 0.2 m a.s.l. At that time, extensive vegetation areas were already affected by drought, even if the spatially averaged water level was still relatively high. During the following months, the subsequent downward trend in EVI slowed down considerably. From September 2018 on, EVI was distinctively higher than normally, indicating an extension of the growing season until late in the year. Mean annual EVI of 0.32 in 2018 compared to the mean of time series 2010–2017 0.27 (std = 0.009) supports this conclusion (Table 1). Interestingly, the drought-induced canopy anomalies became less apparent in the fraction of absorbed photosynthetically active radiation (fPAR). In comparison to EVI, the seasonal dynamics in fPAR formed a broad plateau with maxima up to 0.90, that lasted from May to September. This indicates that there is little variation in the amount of energy absorbed by the canopy during most of the growing season. Further, as the magnitude of fPAR remained constantly high throughout the summer 2018, the drought stress of the vegetation was not reflected by an attenuation of absorbed PAR.

### 3.3  Response of $CO_2$ exchange to drought

The rewetted fen site is highly productive with substantial rates of GEP and $R_{eco}$ (Koebsch et al., 2013). Despite strong interannual variation, the fen has acted as net $CO_2$ sink since rewetting with average NEE budgets of -0.70 kg m$^{-2}$ a$^{-1}$ (Koebsch et al., 2013). New record levels of GEP and $R_{eco}$ were reached in 2018 (Figure 5A and 5B): The annual $R_{eco}$ budget

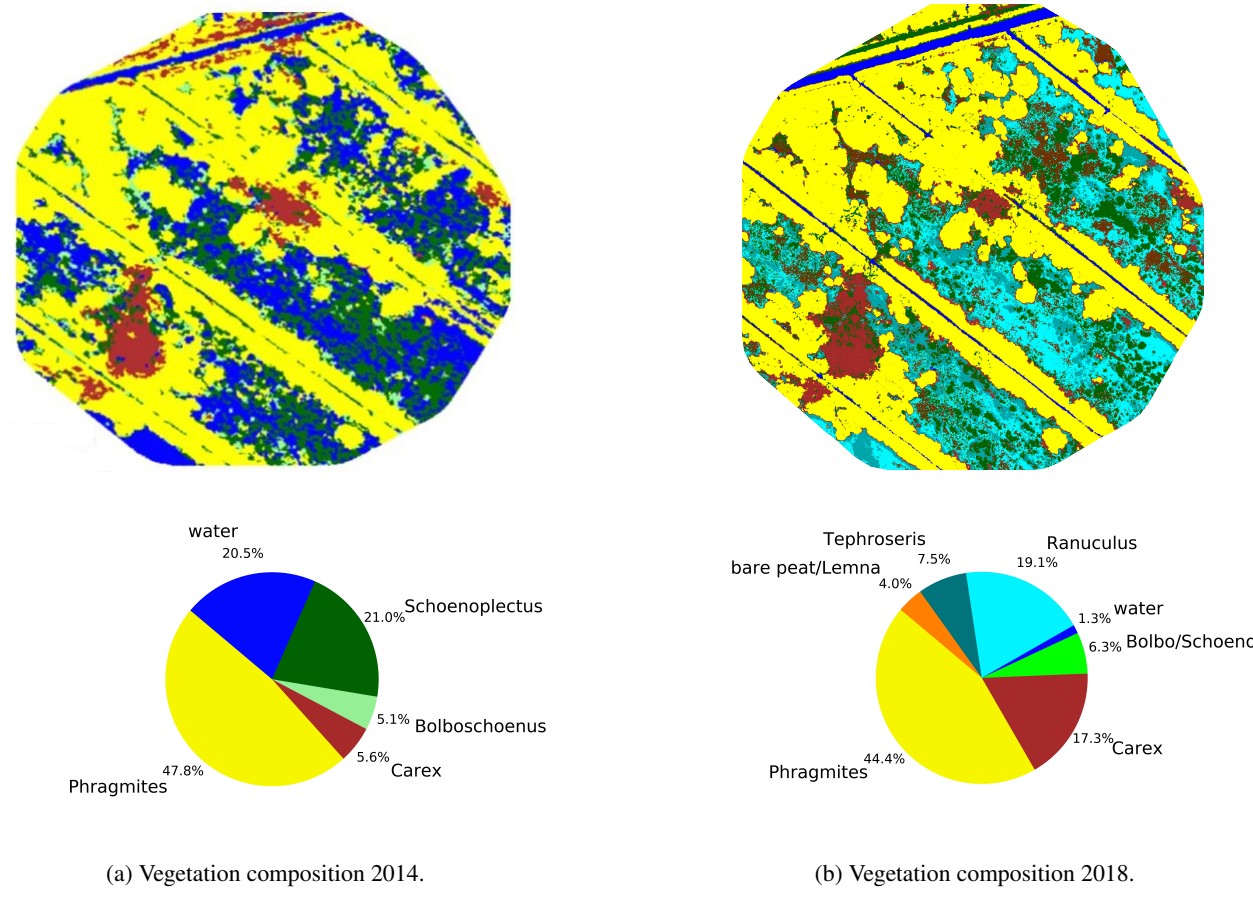

(a) Vegetation composition 2014.      (b) Vegetation composition 2018.

**Figure 4.** Vegetation composition in 2014 (4a) as presented in Koch et al. (2017) and after the drought in 2018 (4b, black border marks study site extend of Koch et al. (2017)).

totalled 3.22 kg $CO_2$ m$^{-2}$ and exceeded the post-rewetting average by 0.93 kg m$^{-2}$. Further, with -3.61 kg $CO_2$ m$^{-2}$ total annual GEP exceeded the average photosynthetic $CO_2$ uptake by 0.63 kg m$^{-2}$. Hence, in 2018 the fen remained a net $CO_2$ sink, though net $CO_2$ sequestration was 0.30 kg m$^{-2}$ lower than in average post-rewetting years.

NEE and its component fluxes marked seasonal dynamics including a decoupling of GEP and R$_{eco}$ when drought took effect from July 2018 on (Figure 5C). Before July, daily R$_{eco}$ and GEP sums were in the upper range of normal years. This is most likely due to high temperatures and radiation supply which fostered efficient growth of the established vegetation. As the rise in C assimilation outweighed the increase in respiratory $CO_2$ release, the first weeks in the growing season 2018 also exhibited comparatively high rates of net $CO_2$ uptake. GEP peaked at -37 g $CO_2$ m$^{-2}$ d$^{-1}$ in June/July which coincided with the maximum EVI. Following this peak, photosynthetic $CO_2$ uptake decreased substantially, which was likely driven by the onset of drought-induced stress for the established vegetation.

This was further supported by the drop in light use efficiency (LUE) of GEP, which halved from 0.18 $g\,C\,MJ^{-1}$ to 0.09 $g\,C\,MJ^{-1}$ between June and July 2018. This drop in LUE was related to a decrease in GEP, i.e., to an attenuation of photosynthetic $CO_2$ uptake, whilst the PAR absorbance characteristics of the canopy remained virtually unaffected. Such a drought-related decrease in LUE has been reported by a variety of peatland studies and is related to stomata closure as common physiological mechanism of vascular plants to cope with water deficit (Connolly et al., 2009; Kross et al., 2016).

At the same time, $R_{eco}$ maintained its upward trend and reached a new record of 25 g $CO_2$ $m^{-2}$ $d^{-1}$ at the end of July. $R_{eco}$ remained on this plateau for the following two months, reflecting a persistent $CO_2$ loss, which is likely to be associated with a shift from prevailing autotrophic to prevailing heterotrophic respiration (Olefeldt et al., 2017). In normal years, the fen smoothly shifts from being a net $CO_2$ sink to a net $CO_2$ source at the end of the growing season. The dry spell in summer 2018, however, caused a rapid switch from net $CO_2$ sink to $CO_2$ neutrality already in July.

After the drought-related decline in July 2018, GEP increased again in August. This 2nd peak in GEP coincided with a sustained upswing in LUE and the observed colonization of dried-up areas by *Tephroseris palustris* and *Ranunculus sceleratus*. LUE reached high values of 0.30 $g\,C\,MJ^{-1}$ even late in the season in October/November. At that time, high rates of photosynthetic $CO_2$ uptake represented by GEP occurred regardless of the decreasing PAR absorbance capacity of the senescencing canopy. *Tephroseris palustris* and *Ranunculus sceleratus* are pioneer plants, the ecophysiology of which is targeted for vigorous biomass production and, thus, efficient $CO_2$ assimilation. Further, GEP rates in autumn 2018 were promoted by unusually high temperatures, that enhance the capacity of photosynthetic $CO_2$ assimilation and increase the maximum photosynthesis rate at light saturation (Lüttge et al., 2010). In accordance, also the CUP 2018 extended until late in the season at day of year (doy) 307. Hence, carbon uptake lasted 26 days longer and extended the length of the total CUP by 33 days in comparison to reference years. Hence, biomass accumulation through the massive spread of pioneer species in combination with high autumn temperatures held GEP rates high until late in the growing season.

### 3.4 Response of $CH_4$ exchange to drought

Annual $CH_4$ sums in the rewetting period 2011–2017 averaged at 66 $g\,m^{-2}$, but fell down to 53 $g\,m^{-2}$ in 2018, which was 20 % below the average of the reference period. The decline in $CH_4$ emissions occurred mainly in the period from August onwards, when daily fluxes kept below 0.2 $g\,CH_4\,m^{-2}\,d^{-1}$ and were thus 60 % lower than in reference years. Preceeding the steep decline in $CH_4$ emissions in August, there was a distinct emission peak with flux rates up to 0.2 $g\,CH_4\,m^{-2}\,d^{-1}$, that occurred when the water table dropped down to surface level. Such a $CH_4$ emission pulse concomitant to falling water tables is commonly associated with degassing due to decreasing hydrostatic pressure (Moore et al., 1990; Dinsmore et al., 2009).

The following drought-induced reduction in $CH_4$ emissions was expected given the shift in the peat redox regime and the adjustments of the methane cycling community. In a complementary study addressing the microbial response to the drought spell, we found a substantial increase in the abundance of type I methanotrophs of the order *Methylococcales* (Unger et al., 2020). Accordingly, the observed reduction in $CH_4$ emissions is most likely due to a combination of inhibited methanogenesis under the presence of oxygen and other terminal electron appectors and an increase in microbial $CH_4$ consumption.

$N_2O$ is another effective and long-lived greenhouse gas of potential relevance in peatlands. $N_2O$ is produced from incomplete turnover reaction of organic nitrogen compounds (Bremner and Blackmer, 1980) and can substantially contribute to the

315 radiative forcing of drained peatlands (Günther et al., 2020). However, as emissions cease under the anaerobic conditions, $N_2O$ is not of primary concern for most rewetted peatlands (Hendriks et al., 2007). The full greenhouse gas balance of an abandoned peat meadow). Indeed, our own flux measurements conducted at the study site in the year prior to rewetting in 2009 indicated $N_2O$ emissions to be negligible (Koebsch, 2009). Yet, we cannot exclude, that the alternating water tables occurring in summer 2018 can stimulate $N_2O$ production and thereby add to the radiative forcing of peatlands affected by drought.

## 4 Drought response mechanisms of restored fens

Peatland conservation and rewetting is considered one of the major natural climate solutions (Griscom et al., 2017; Leifeld and Menichetti, 2018). In comparison to afforestation in monoculture plantations, peatland protection is expected to conserve or recreate self-regulating ecosystems that are resilient to climate impacts (Leifeld and Menichetti, 2018; Seddon et al., 2020). Nevertheless, in view of increasing frequency and severity of climatic extreme events (Pachauri et al., 2014), the effects of

325 temporary droughts on the functioning of rewetted peatlands are still largely unexplored and lead to considerable uncertainty with regard to the inherent climate mitigation goals.

Pristine peatlands are adaptive systems characterized by quasi-stable equilibrium states and feature resilience mechanisms to cope with drought to a certain extent (Dise, 2009). The ecohydrology of intact peat is characterized by its large water holding capacity and its capillary wicking processes (Ingram, 1987; Lapen et al., 2000). Whilst these present efficient regulation mecha-

330 nism to buffer short-term dry spells, persistent drought or increasing drought frequency can also induce shifts in vegetation and C regime (Couwenberg and Joosten, 1999; Couwenberg et al., 2008). In mires, drought can induce changes from low-phenolic Sphagnum/herbs towards phenol-rich shrub vegetation which increases C sequestration and protects soil C (Riutta et al., 2007; Limpens et al., 2008; Wang et al., 2015). Drought can even trigger abrupt episodes of habitat conversion, which are essential for the succession trajectory of peatlands. Such drought-induced state-shifts are known for kettle peatland development and are

335 associated with greatly increased C accumulation rates (Ireland et al., 2012).

Analogue climate-feedback mechanisms cannot be anticipated for degraded restored fens, where catchment hydrology, soil and trophic conditions as well as propagule availability have been subject to irreversible change (van Diggelen et al., 2006; Klimkowska et al., 2010). Here, we describe a distinct response mechanism of such newly created systems to severe drought: Sinking water levels exposed bare spots, that were rapidly colonized by pioneer species. Hence, after years of stagnant vegeta-

340 tion development, drought acted as a trigger event to close persistent vegetation gaps. Our study shows, how drought-induced founding effects can give impetus to overcome stagnant vegetation succession of rewetted fens, the canopies of which are often interspersed by more or less extended open water patches where vegetation cannot take root (Steffenhagen et al., 2012; Matthes et al., 2014; Franz et al., 2016). During the build-up of new biomass, substantial amounts of $CO_2$ were sequestered which overcompensated for the drought-induced decline of photosynthetic $CO_2$ uptake by the established vegetation. On an

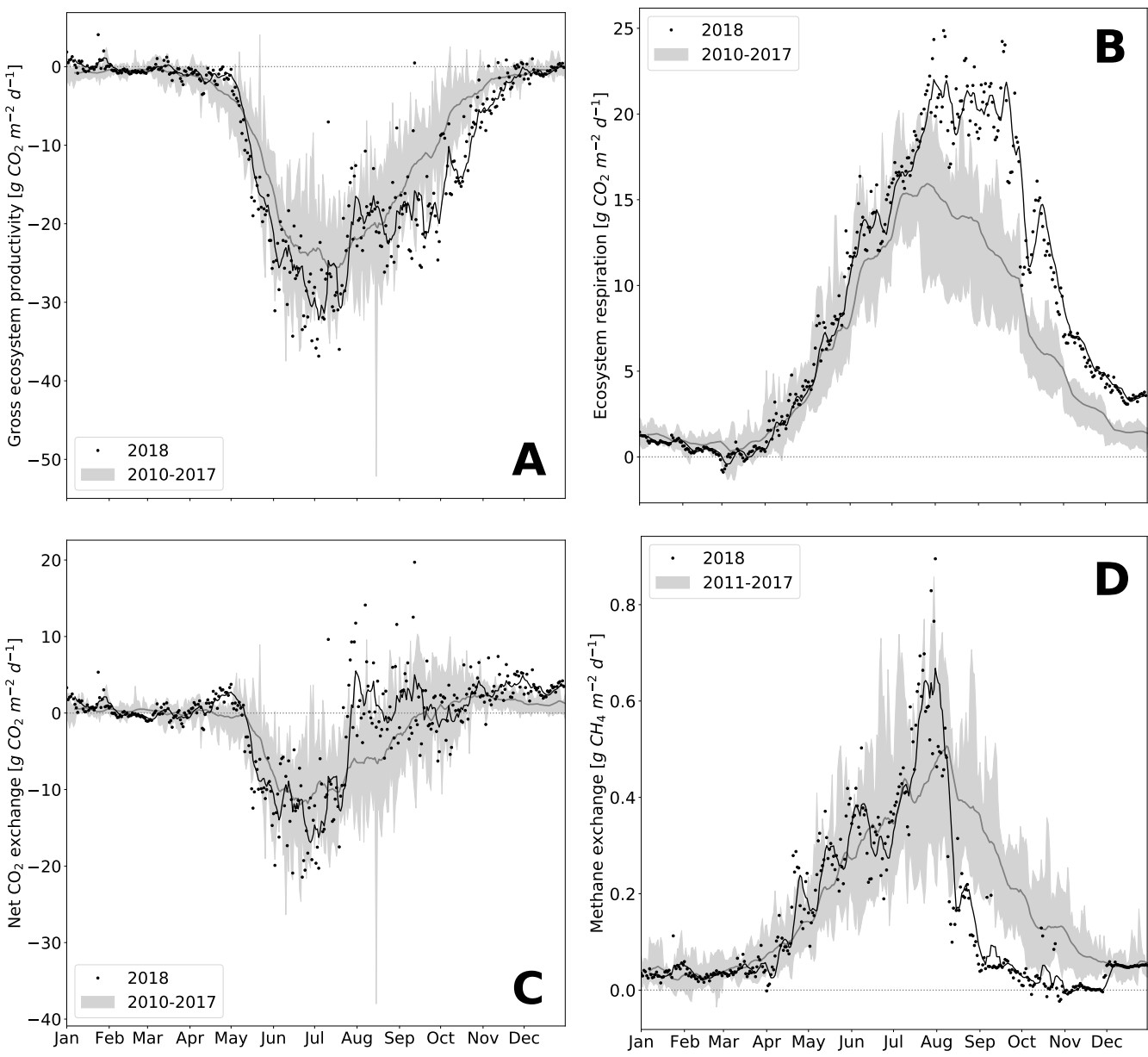

**Figure 5.** Component fluxes GEP (A) and $R_{eco}$ (B) of NEE (C) and $CH_4$ (D) over the course of the year. Variables are represented as black line (7 days rolling mean of black dots) for 2018 whereas the grey shading represents the variable range (minimum-maximum) throughout the reference period 2010–2017 (dark grey line is the mean of the reference period).

annual basis, enhanced GEP offset half of the drought-induced increase in $R_{eco}$. Therefore, the restored fen maintained its net $CO_2$ sink function even in such a year of extreme drought.

The rapid colonization by pioneer species and the associated $CO_2$ uptake during the peak of the drought in August 2018 was only possible because there was still sufficient moisture for germination. When rainfalls stopped in May, the water reservoirs in the fen under study were well filled, which dampened the severity of the drought. Such buffer properties result from the hydrological sink function characteristic for fens which are commonly fed by various inflows. Therefore, the mechanisms described above cannot be transferred to raised bogs, which are exclusively fed by precipitation and are likely to be affected by drought to a greater extent (Dise, 2009). Overall, our study suggests, that chances of restoring self-regulating fens under increasing frequency and severity of droughts improve if the peatland can regain its natural function as hydrological sink which, in turn, depends on the hydrological connectivity still existing in the catchment.

The reduction of $CH_4$ emissions under low water tables is quite common, and this fact is also used to reduce $CH_4$ emissions from rice cultivation through the deliberate introduction of periodic drought (Runkle et al., 2019). $CH_4$ emissions cause a substantial radiative forcing peak in the first decades of peatland rewetting (Günther et al., 2020). Therefore, active water management for the temporary introduction of aerobic conditions could also be considered to optimize the mitigation potential of peatland rewetting as nature-based climate solutions (Unger et al., 2020). Nevertheless, such measures must be assessed with regard to their impact on other ecosystem functions and weighed against possible effects on $CO_2$ and $N_2O$ exchange.

As much as the immediate effects of temporary droughts are important, it is conceivable, that such extreme events initiate distinct carry-over effects that extend beyond the actual drought period and can set the course for the future development of restored fens and their C cycle. Though, in practice, it is difficult to unravel such aftereffects of past events from contemporary influences. For example, we could still observe the presence of *Tephroseris palustris*, despite the resuming water level rise in the year after the drought. However since the majority of the resupplied water originated from an episodic brackish water intrusion event in January 2019, we cannot generalize the observations from 2019 to common freshwater fens. Since our own data are not suited to address the post-drought development under common hydrological conditions, we provide some considerations for possible future scenarios for fens affected by drought:

1. The relevance of drought-induced founding events for the long-term succession of restored fens will rely on the capability of the newly formed vegetation to gain a lasting foothold in these systems. Dependent on whether these pioneer species can cope with the recurrent water level rise (Koch et al., 2017), they will contribute to the ecosystems C budget in one way or the other: If the drought event can indeed accelerate the closure of persistent canopy gaps, it could increase photosynthetic $CO_2$ sequestration and C accumulation in the long run. A comparison to another drought-affected fen has shown that the chances of the new vegetation to gain a foothold in the long term increase, if the founding event includes species that already predominate on the site (Koebsch et al., 2020). However, if, the new vegetation declines after the return of normal hydrological conditions, the dead biomass will form a large pool of easily decomposable C. Eventually, this C will be released as $CO_2$ and $CH_4$, so that the radiative forcing effect of drought could simply be postponed to the following years. Still, even in this unfavorable case, the die-back of the new vegetation could initiate silting processes in flooded peatlands and thereby set the stage for subsequent peat-forming vegetation.

2. While the potential die-back of the newly formed vegetation could feed $CH_4$ production in the post-drought period, existing research indicates alternative scenarios in which drought alters the redox geochemistry of peat to sustainably reduce $CH_4$ emissions. For example, falling water tables can recharge the stock of electron acceptors, thereby establishing thermodynamically unfavorable conditions for methanogenesis (Knorr and Blodau, 2009). Furthermore, drought can affect the methane cycling community by increasing the abundance of methanotrophs and/or declining the abundance of methanogens (Unger et al., 2020). In either of these cases, the temporal suspension of $CH_4$ emissions beyond the actual drought period would contribute to improve the climate balance of peatland rewetting.

In view of the divergent succession trajectories and the contrasting climate mitigation prospects for peatlands affected by drought, there is substantial demand for ecosystem-scale studies to delineate drought impacts in relation to climate-normal years and, further, to track the post-drought development of the site under consideration. In this respect, our study provides a starting point to demonstrate the far-reaching implications of drought events under special consideration of the link between vegetation response and greenhouse gas exchange. Although designed as a case study, we believe that our observations are transferable to a wider range of degraded, rewetted fens, as many of these sites are resembling each other in terms of hydrology and canopy characteristics. Further research is of particular relevance given the role of peatland rewetting in nature-based climate solutions and the need to meet the mitigation expectations under a changing climate.

*Code availability.* Both, the classification script and the script to calculate spectral indices can be found at github.com/florianbeyer/RandomForest-Classification and github.com/florianbeyer/SpectralIndices.

## Appendix A: ORCID IDs

| | |
|---|---|
| Florian Beyer: | 0000-0002-9203-320X |
| Franziska Koebsch: | 0000-0003-1045-7680 |
| Gerald Jursinski: | 0000-0002-6248-9388 |
| Florian Jansen: | 0000-0002-0331-5185 |

# Appendix B: Supplementary material

 **B1   UAS data sets**

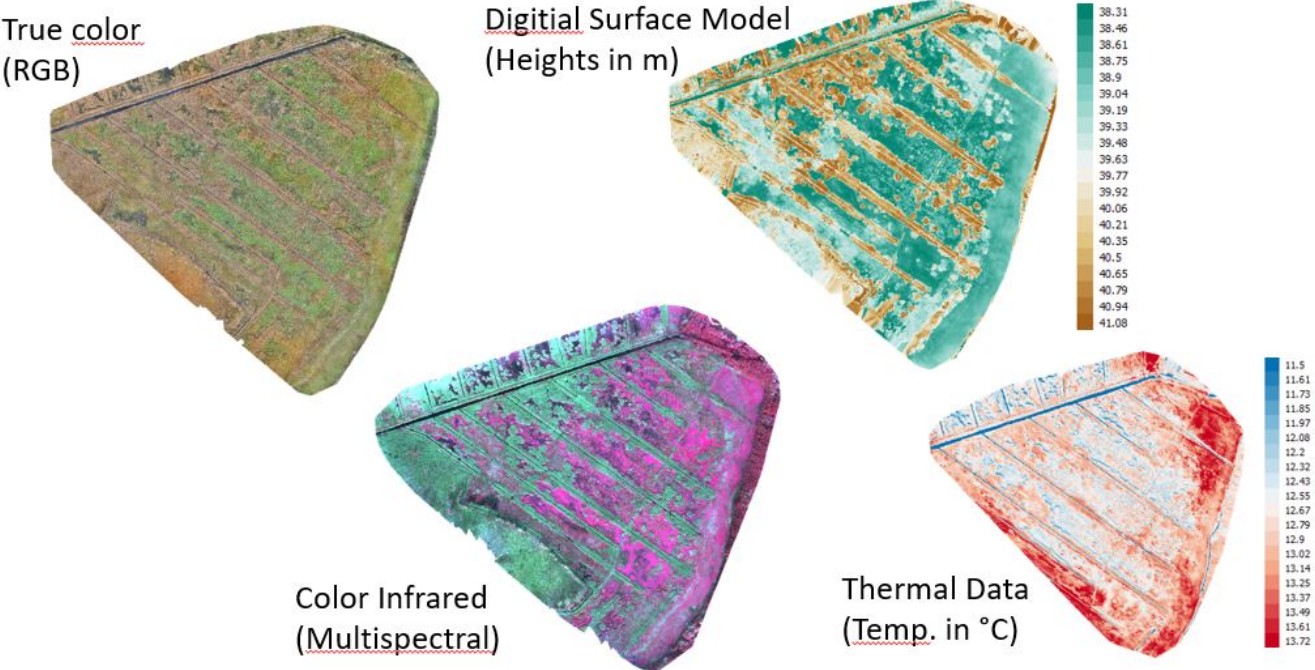

**Figure B1.** True color, multispectral (band combination: near infrared|red|green), digital surface model and thermal orthomosaic of the multisensor UAS data.

## B2 Modis footprint

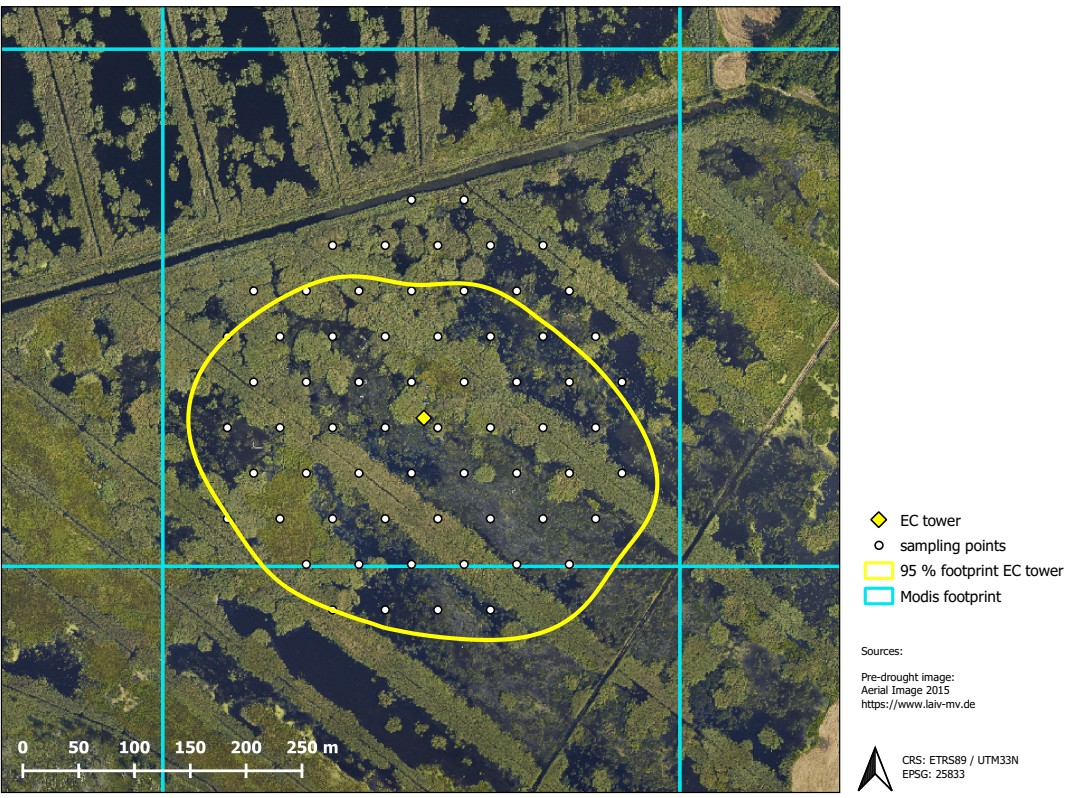

**Figure B2.** Spatial coverage of the different data sources including the 95 % footprint climatology of the eddy covariance (EC) flux, ground truthing points for vegetation mapping, and the grid cell used for MODIS vegetation indices.

## B3 Bands of the Multisensor data set and its importances for classification

**Table B1.** Multisensor data set consits of 107 bands. All indices are described in github.com/florianbeyer/SpectralIndices.

| No. | Band name | Type/Meaning | Data from | Derived from | No. | Band name | Type/Meaning | Data from | Derived from |
|-----|-----------|--------------|-----------|--------------|-----|-----------|--------------|-----------|--------------|
| 1 | RGB1 | Blue | RGB Sensor | | 55 | fe3 | Spectral Index | | Multispectral Sensor |
| 2 | RGB2 | Green | RGB Sensor | | 56 | gemi | Spectral Index | | Multispectral Sensor |
| 3 | RGB3 | Red | RGB Sensor | | 57 | gndvi | Spectral Index | | Multispectral Sensor |
| 4 | MS1 | Green | Multispectral Sensor | | 58 | osavi1 | Spectral Index | | Multispectral Sensor |
| 5 | MS2 | Red | Multispectral Sensor | | 59 | osavi2 | Spectral Index | | Multispectral Sensor |
| 6 | MS3 | Red Edge | Multispectral Sensor | | 60 | pvr | Spectral Index | | Multispectral Sensor |
| 7 | MS4 | Near Infrared | Multispectral Sensor | | 61 | rdvi | Spectral Index | | Multispectral Sensor |
| 8 | DSM | Digital Surface Model (DSM) | | RGB Sensor | 62 | rededge2 | Spectral Index | | Multispectral Sensor |
| 9 | th_index | Thermal | Thermal Sensor | | 63 | savi | Spectral Index | | Multispectral Sensor |
| 10 | ngrdi | Spectral Index | | RGB Sensor | 64 | sbl | Spectral Index | | Multispectral Sensor |
| 11 | tgi | Spectral Index | | RGB Sensor | 65 | spvi | Spectral Index | | Multispectral Sensor |
| 12 | vari | Spectral Index | | RGB Sensor | 66 | tc_gvimss | Spectral Index | | Multispectral Sensor |
| 13 | exg | Spectral Index | | RGB Sensor | 67 | tc_nsimss | Spectral Index | | Multispectral Sensor |
| 14 | gcc | Spectral Index | | RGB Sensor | 68 | tc_sbimss | Spectral Index | | Multispectral Sensor |
| 15 | gli | Spectral Index | | RGB Sensor | 69 | tc_yvimss | Spectral Index | | Multispectral Sensor |
| 16 | ari | Spectral Index | | Multispectral Sensor | 70 | tcari | Spectral Index | | Multispectral Sensor |
| 17 | arvi2 | Spectral Index | | Multispectral Sensor | 71 | tcari_osavi | Spectral Index | | Multispectral Sensor |
| 18 | atsavi | Spectral Index | | Multispectral Sensor | 72 | tcari2 | Spectral Index | | Multispectral Sensor |
| 19 | avi | Spectral Index | | Multispectral Sensor | 73 | tci | Spectral Index | | Multispectral Sensor |
| 20 | bri | Spectral Index | | Multispectral Sensor | 74 | tvi | Spectral Index | | Multispectral Sensor |
| 21 | ccci | Spectral Index | | Multispectral Sensor | 75 | varirededge | Spectral Index | | Multispectral Sensor |
| 22 | chlgreen | Spectral Index | | Multispectral Sensor | 76 | wdrvi | Spectral Index | | Multispectral Sensor |
| 23 | chlrededge | Spectral Index | | Multispectral Sensor | 77 | ndrdi | Spectral Index | | Multispectral Sensor |
| 24 | cigreen | Spectral Index | | Multispectral Sensor | 78 | ndre | Spectral Index | | Multispectral Sensor |
| 25 | cirededge | Spectral Index | | Multispectral Sensor | 79 | ndvi | Spectral Index | | Multispectral Sensor |
| 26 | ctvi | Spectral Index | | Multispectral Sensor | 80 | nli | Spectral Index | | Multispectral Sensor |
| 27 | cvi | Spectral Index | | Multispectral Sensor | 81 | normg | Spectral Index | | Multispectral Sensor |
| 28 | datt1 | Spectral Index | | Multispectral Sensor | 82 | normnir | Spectral Index | | Multispectral Sensor |
| 29 | datt4 | Spectral Index | | Multispectral Sensor | 83 | normr | Spectral Index | | Multispectral Sensor |
| 30 | ddn | Spectral Index | | Multispectral Sensor | 84 | band1_Energy | Texture Index | | RGB Sensor |
| 31 | diff1 | Spectral Index | | Multispectral Sensor | 85 | band1_Entropy | Texture Index | | RGB Sensor |
| 32 | diff2 | Spectral Index | | Multispectral Sensor | 86 | band1_Correlation | Texture Index | | RGB Sensor |
| 33 | dvimss | Spectral Index | | Multispectral Sensor | 87 | band1_InverseDifferenceMoment | Texture Index | | RGB Sensor |
| 34 | gosavi | Spectral Index | | Multispectral Sensor | 88 | band1_Inertia | Texture Index | | RGB Sensor |
| 35 | grndvi | Spectral Index | | Multispectral Sensor | 89 | band1_ClusterShade | Texture Index | | RGB Sensor |
| 36 | lai | Spectral Index | | Multispectral Sensor | 90 | band1_ClusterProminence | Texture Index | | RGB Sensor |
| 37 | lci | Spectral Index | | Multispectral Sensor | 91 | band1_HaralickCorrelation | Texture Index | | RGB Sensor |
| 38 | logr | Spectral Index | | Multispectral Sensor | 92 | band2_Energy | Texture Index | | RGB Sensor |
| 39 | maccioni | Spectral Index | | Multispectral Sensor | 93 | band2_Entropy | Texture Index | | RGB Sensor |
| 40 | mari | Spectral Index | | Multispectral Sensor | 94 | band2_Correlation | Texture Index | | RGB Sensor |
| 41 | mcari | Spectral Index | | Multispectral Sensor | 95 | band2_InverseDifferenceMoment | Texture Index | | RGB Sensor |
| 42 | mcari_mtvi2 | Spectral Index | | Multispectral Sensor | 96 | band2_Inertia | Texture Index | | RGB Sensor |
| 43 | mcari_osavi | Spectral Index | | Multispectral Sensor | 97 | band2_ClusterShade | Texture Index | | RGB Sensor |
| 44 | mcari1 | Spectral Index | | Multispectral Sensor | 98 | band2_ClusterProminence | Texture Index | | RGB Sensor |
| 45 | mcari2 | Spectral Index | | Multispectral Sensor | 99 | band2_HaralickCorrelation | Texture Index | | RGB Sensor |
| 46 | mgvi | Spectral Index | | Multispectral Sensor | 100 | band3_Energy | Texture Index | | RGB Sensor |
| 47 | mnsi | Spectral Index | | Multispectral Sensor | 101 | band3_Entropy | Texture Index | | RGB Sensor |
| 48 | msavi | Spectral Index | | Multispectral Sensor | 102 | band3_Correlation | Texture Index | | RGB Sensor |
| 49 | msbi | Spectral Index | | Multispectral Sensor | 103 | band3_InverseDifferenceMoment | Texture Index | | RGB Sensor |
| 50 | msr670 | Spectral Index | | Multispectral Sensor | 104 | band3_Inertia | Texture Index | | RGB Sensor |
| 51 | mtvi2 | Spectral Index | | Multispectral Sensor | 105 | band3_ClusterShade | Texture Index | | RGB Sensor |
| 52 | myvi | Spectral Index | | Multispectral Sensor | 106 | band3_ClusterProminence | Texture Index | | RGB Sensor |
| 53 | evi2 | Spectral Index | | Multispectral Sensor | 107 | band3_HaralickCorrelation | Texture Index | | RGB Sensor |
| 54 | evi22 | Spectral Index | | Multispectral Sensor | | | | | |

**Table B2.** All bands of the multisensor data set orderd by the GINI coefficient. The higher the GINI the more important is the band for the Random Forest classification.

| No. | Band | Gini | Gini (%) | c. Gini | No. | Band | Gini | Gini (%) | cumulative Gini |
|---|---|---|---|---|---|---|---|---|---|
| 8 | DSM | 0.06415 | 6.4 | 6.4 | 63 | savi | 0.00618 | 0.6 | 85.0 |
| 35 | grndvi | 0.03760 | 3.8 | 10.2 | 100 | band3_Energy | 0.00596 | 0.6 | 85.6 |
| 82 | normnir | 0.03268 | 3.3 | 13.4 | 85 | band1_Entropy | 0.00560 | 0.6 | 86.1 |
| 17 | arvi2 | 0.02773 | 2.8 | 16.2 | 106 | band3_ClusterProminence | 0.00556 | 0.6 | 86.7 |
| 50 | msr670 | 0.02674 | 2.7 | 18.9 | 96 | band2_Inertia | 0.00551 | 0.6 | 87.2 |
| 74 | tvi | 0.02510 | 2.5 | 21.4 | 45 | mcari2 | 0.00539 | 0.5 | 87.8 |
| 38 | logr | 0.02499 | 2.5 | 23.9 | 87 | band1_InverseDifferenceMoment | 0.00522 | 0.5 | 88.3 |
| 76 | wdrvi | 0.02460 | 2.5 | 26.4 | 43 | mcari_osavi | 0.00522 | 0.5 | 88.8 |
| 52 | myvi | 0.02302 | 2.3 | 28.7 | 93 | band2_Entropy | 0.00517 | 0.5 | 89.3 |
| 49 | msbi | 0.02271 | 2.3 | 30.9 | 54 | evi22 | 0.00498 | 0.5 | 89.8 |
| 40 | mari | 0.02140 | 2.1 | 33.1 | 95 | band2_InverseDifferenceMoment | 0.00492 | 0.5 | 90.3 |
| 30 | ddn | 0.02102 | 2.1 | 35.2 | 48 | msavi | 0.00486 | 0.5 | 90.8 |
| 5 | MS2 | 0.02093 | 2.1 | 37.3 | 80 | nli | 0.00485 | 0.5 | 91.3 |
| 79 | ndvi | 0.02086 | 2.1 | 39.4 | 102 | band3_Correlation | 0.00485 | 0.5 | 91.8 |
| 26 | ctvi | 0.01867 | 1.9 | 41.2 | 53 | evi2 | 0.00478 | 0.5 | 92.3 |
| 34 | gosavi | 0.01826 | 1.8 | 43.0 | 101 | band3_Entropy | 0.00477 | 0.5 | 92.7 |
| 67 | tc_nsimss | 0.01819 | 1.8 | 44.9 | 84 | band1_Energy | 0.00456 | 0.5 | 93.2 |
| 64 | sbl | 0.01775 | 1.8 | 46.6 | 66 | tc_gvimss | 0.00443 | 0.4 | 93.6 |
| 83 | normr | 0.01750 | 1.7 | 48.4 | 29 | datt4 | 0.00435 | 0.4 | 94.1 |
| 47 | mnsi | 0.01665 | 1.7 | 50.1 | 36 | lai | 0.00432 | 0.4 | 94.5 |
| 31 | diff1 | 0.01630 | 1.6 | 51.7 | 44 | mcari1 | 0.00432 | 0.4 | 94.9 |
| 68 | tc_sbimss | 0.01529 | 1.5 | 53.2 | 81 | normg | 0.00386 | 0.4 | 95.3 |
| 75 | varirededge | 0.01527 | 1.5 | 54.7 | 104 | band3_Inertia | 0.00370 | 0.4 | 95.7 |
| 70 | tcari | 0.01515 | 1.5 | 56.3 | 65 | spvi | 0.00367 | 0.4 | 96.0 |
| 7 | MS4 | 0.01454 | 1.5 | 57.7 | 11 | tgi | 0.00301 | 0.3 | 96.3 |
| 22 | chlgreen | 0.01404 | 1.4 | 59.1 | 98 | band2_ClusterProminence | 0.00291 | 0.3 | 96.6 |
| 60 | pvr | 0.01399 | 1.4 | 60.5 | 4 | MS1 | 0.00289 | 0.3 | 96.9 |
| 6 | MS3 | 0.01375 | 1.4 | 61.9 | 105 | band3_ClusterShade | 0.00275 | 0.3 | 97.2 |
| 55 | fe3 | 0.01319 | 1.3 | 63.2 | 90 | band1_ClusterProminence | 0.00268 | 0.3 | 97.5 |
| 33 | dvimss | 0.01283 | 1.3 | 64.5 | 32 | diff2 | 0.00257 | 0.3 | 97.7 |
| 24 | cigreen | 0.01272 | 1.3 | 65.8 | 14 | gcc | 0.00246 | 0.2 | 98.0 |
| 19 | avi | 0.01267 | 1.3 | 67.0 | 15 | gli | 0.00240 | 0.2 | 98.2 |
| 9 | th_index | 0.01096 | 1.1 | 68.1 | 89 | band1_ClusterShade | 0.00222 | 0.2 | 98.4 |
| 27 | cvi | 0.01083 | 1.1 | 69.2 | 59 | osavi2 | 0.00169 | 0.2 | 98.6 |
| 57 | gndvi | 0.00977 | 1.0 | 70.2 | 10 | ngrdi | 0.00160 | 0.2 | 98.8 |
| 71 | tcari_osavi | 0.00975 | 1.0 | 71.2 | 86 | band1_Correlation | 0.00145 | 0.1 | 98.9 |
| 77 | ndrdi | 0.00957 | 1.0 | 72.1 | 12 | vari | 0.00144 | 0.1 | 99.1 |
| 107 | band3_HaralickCorrelation | 0.00896 | 0.9 | 73.0 | 69 | tc_yvimss | 0.00142 | 0.1 | 99.2 |
| 58 | osavi1 | 0.00885 | 0.9 | 73.9 | 94 | band2_Correlation | 0.00097 | 0.1 | 99.3 |
| 56 | gemi | 0.00879 | 0.9 | 74.8 | 97 | band2_ClusterShade | 0.00085 | 0.1 | 99.4 |
| 91 | band1_HaralickCorrelation | 0.00874 | 0.9 | 75.7 | 78 | ndre | 0.00070 | 0.1 | 99.4 |
| 103 | band3_InverseDifferenceMoment | 0.00838 | 0.8 | 76.5 | 25 | cirededge | 0.00062 | 0.1 | 99.5 |
| 73 | tci | 0.00816 | 0.8 | 77.3 | 23 | chlrededge | 0.00059 | 0.1 | 99.6 |
| 16 | ari | 0.00811 | 0.8 | 78.1 | 72 | tcari2 | 0.00058 | 0.1 | 99.6 |
| 18 | atsavi | 0.00773 | 0.8 | 78.9 | 21 | ccci | 0.00051 | 0.1 | 99.7 |
| 99 | band2_HaralickCorrelation | 0.00772 | 0.8 | 79.7 | 61 | rdvi | 0.00051 | 0.1 | 99.7 |
| 51 | mtvi2 | 0.00727 | 0.7 | 80.4 | 37 | lci | 0.00051 | 0.1 | 99.8 |
| 20 | bri | 0.00686 | 0.7 | 81.1 | 62 | rededge2 | 0.00050 | 0.1 | 99.8 |
| 42 | mcari_mtvi2 | 0.00682 | 0.7 | 81.8 | 28 | datt1 | 0.00046 | 0.0 | 99.9 |
| 88 | band1_Inertia | 0.00660 | 0.7 | 82.4 | 39 | maccioni | 0.00038 | 0.0 | 99.9 |
| 92 | band2_Energy | 0.00659 | 0.7 | 83.1 | 3 | RGB3 | 0.00035 | 0.0 | 100.0 |
| 41 | mcari | 0.00643 | 0.6 | 83.7 | 13 | exg | 0.00029 | 0.0 | 100.0 |
| 46 | mgvi | 0.00627 | 0.6 | 84.3 | 1 | RGB1 | 0.00011 | 0.0 | 100.0 |
| | | | | | 2 | RGB2 | 0.00009 | 0.0 | 100.0 |

*Author contributions.* Franziska Koebsch conceived the study. Florian Beyer and Franziska Koebsch carried out the experiments and wrote the manuscript. Florian Jansen and Gerald Jurasinski revised the manuscript and contributed with helpful comments. Marian Koch revised the manuscript and conducted the first studies on which the manuscript is based. Birgit Schröder carried out the vegetation mapping and helped with botanical issues.

*Competing interests.* The authors declare that they have no conflicts of interest.

*Acknowledgements.* This paper is dedicated to the memory of our dear colleague Dr. Marian Koch, whose work formed the basis for this study. Marian sadly passed away during the final writing phase of this paper. We thank J. Harmuth and A. Stoll from the local administration of forest conservation for granting access to the study site. FB and FK were funded by the European Social Fund (ESF) and the Ministry of Education, Science and Culture of Mecklenburg-Western Pomerania within the scope of the project WETSCAPES (ESF/14-BM-A55-0034/16). This is Baltic TRANSCOAST publication xxxx. The Helmholtz Terrestrial Environmental Observatories (TERENO) Network supported the long-term operation of the eddy covariance measurements. Special gratitude is owed to J. Hofmann for his tireless commitment to field work under harsh conditions.

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
