# Peer review of "Drought years in peatland rewetting: Rapid vegetation succession can maintain the net $CO_2$ sink function"

_Biogeosciences, 2020_

## Referee Comment (RC1) · Anonymous Referee #1 · 10 Aug 2020

Peer Review: Florian Beyer et al. Submitted, EGU Biogeoscience August 2020

Thanks for the opportunity to review your interesting manuscript. This work delivers a snapshot of the biogeochemical impact of drought processes on restored peatlands, using a variety of remote sensing and micrometeorological methods. Given the potential importance of restored peatlands as a 'natural climate solution', it is critical to understand the potential carbon cycle changes associated with disturbances, such as prolonged drought.

The manuscript's strengths lie in laying out the multi-year data – which essentially constitutes a comparison between previous 'normal' years, and the 2018 disturbance

year. However, it is light on context, discussion, and implications. Below are some questions that may be considered, that would add more richness to the manuscript.

General comments:

1. Consider adding more context, in the intro, concerning why re-wetted peatlands could be an important climate change mitigation strategy. One important consideration is that many 'natural climate solution' potential portfolios often lack any consideration for how future climate change and disturbance regimes will impact the potential enhanced (or avoided) sequestration. There is an opportunity to better make the case of how crucial it is to use natural experiments like this to understand the implications of disturbance on C sink potential of restored landscapes. Some references on NCS and peatlands:

Leifeld, J., & Menichetti, L. (2018). The underappreciated potential of peatlands in global climate change mitigation strategies. Nature Communications, 9(1), 1071. https://doi.org/10.1038/s41467-018-03406-6

Griscom, B. W., Adams, J., Ellise, P. W., Houghton, R. A., & Lomax, G. (2017). Natural Climate Solutions. Proceedings of the National Academy of Sciences, (6), 11–12. https://doi.org/10.1073/pnas.1710465114

Bossio, D. A., Ellis, P. W., Fargione, J., Sanderman, J., Smith, P., Wood, S., et al. (2020). The role of soil carbon in natural climate solutions. Nature Sustainability, 0–1. https://doi.org/10.1038/s41893-020-0491-z

2. Get more explicit about methane and N2O, two other important GHG's that are quite dynamic in wetland/peatland systems, especially during drawdowns.

While I understand the focus on CO2, given the long-term climate forcing of re-wetted peatlands (supported by Gunther et al, 2020), the paper seems to drop the fact that drought, and thus water table dropping, will have effects on denitrification and N2O evolution, as well as redox conditions and CH4 evolution. This needs to be addressed.

[Figure]

For example (different ecosystem), but some rice growers have attempted alternate wetting and drying to reduce CH4 in rice production.

Runkle, B. R. K., Suvočarev, K., Reba, M. L., Reavis, C. W., Smith, S. F., Chiu, Y. L., & Fong, B. (2019). Methane Emission Reductions from the Alternate Wetting and Drying of Rice Fields Detected Using the Eddy Covariance Method. Environmental Science and Technology, 53(2), 671–681. https://doi.org/10.1021/acs.est.8b05535

3. More discussion of future legacy effects. As you say in the final lines, "Although our observations are confined to the year of drought, it is conceivable, that such extreme events initiate distinct carry-over effects that extend beyond the actual drought period and can set the course for the future development of restored fens and their C cycle". As much as the immediate impacts are relevant, it seems like the legacy effects of post-disturbance are as interesting, biogeochemically. Is it possible to include 2019, to say something about these impacts? What is the difference between a long and short disturbance? Between intense and gradual disturbances? Are there any other disturbance types in the record at your site? It could be helpful to at least explicitly describe 2-3 hypotheses of how the legacy effects of disturbance will impact future multi-year biogeochemistry. If the initial disturbance resulted in a reduced, but persistent, CO2 sink, but the following year when water rises again the system 'crashes', this would change the story substantially. How can we manage restored peatlands for future disturbances?

4. Better diagnosis of the mechanistic biophysical drivers of the biogeochemical changes. It is unclear what the specific biophysical cause of the reduced GEP and enhanced Reco is during the drought, and it would be interesting, especially if we are to integrate this disturbance response information into broader models, to diagnose or at least discuss. Right now, 'drought' constitutes less precip, dropping water table, and higher temps. Can you use variation in other years of observations, in which one of these parameters changed and the others were relatively constant, to tease apart what is actually driving this?

[Figure]

One approach is an information theory approach that can help distinguish between drivers, and lags, associated with temp vs. precip vs. water table vs. vpd impacts on the CO2 sink function. See, for CH4, for example:

Sturtevant, C., Ruddell, B. L., Knox, S. H., Verfaillie, J. G., Matthes, J. H., Oikawa, P. Y., & Baldocchi, D. D. (2016). Identifying scale-emergent, nonlinear, asynchronous processes of wetland methane exchange. Journal of Geophysical Research: Biogeosciences, 121(1), 188–204. https://doi.org/10.1002/2015JG003054

Specific comments:

Line 27: "Therefore, climate mitigation measures in peatlands need to focus primarily on the reduction of the CO2 source". Though isn't reducing CH4 of the resulting rewetted peatland also a goal, and could lessen the short-term warming?

Line 37: "drought implies a lowering of the ground water level" What about impacts on CH4 evolution if redox conditions change? Lowering of the ground water level should oxidize and cause less CH4 emission. See this study on the impacts of drought-induced salinization on restored wetlands in California, using mutual information approach :

Chamberlain, S. D., Hemes, K. S., Eichelmann, E., Szutu, D. J., Verfaillie, J. G., & Baldocchi, D. D. (2019). Effect of Drought-Induced Salinization on Wetland Methane Emissions, Gross Ecosystem Productivity, and Their Interactions. Ecosystems, 1–14. https://doi.org/10.1007/s10021-019-00430-5

Technical corrections:

abstract, line 21: 'even by' needs to be reworded to be clearer line 35: remove 'also'

---

## Referee Comment (RC2) · Anonymous Referee #2 · 13 Aug 2020

Beyer et al. analyse the impact of a drought year on the CO2 fluxes of a rewetted fen. They test if drought can hinder the success of peatland rewetting and find that even in a drought year the net CO2 function of the fen can be maintained. They suggest that the increased GPP related to vegetation encroachment in previously open water surfaces compensates partly for the increase in ecosystem respiration.

The manuscript aims to improve our current understanding of vegetation succession following rewetting of fens and specifically of the role of droughts in the success of rewetting efforts. The authors present an interesting mechanism that could explain how rewetted fens respond to droughts. The manuscript is well written, and the authors use

a range of different methodologies to tackle their research question. However, I have a few comments that the authors should consider addressing.

I am wondering if the results of this study can be generalised. The authors present an interesting case study, but it could be that specific site characteristics and the specific drought characteristics were mainly responsible for the observed responses in CO2 fluxes. The authors mention that the water table in spring 2018 was unusually high due to the previous year's high precipitation. This apparently led to a discrepancy between meteorological drought (mainly in May) and hydrological drought (from August on when the water table dropped below the surface). This specific setting might be responsible for the high GPP rates early in the growing season (see Fig. 5a). If the meteorological and hydrological drought would co-occur, a negative effect on GPP could be possible potentially leading to the fen becoming a net CO2 source. This hypothesis might be difficult to test with the existing data, but the authors might consider discussing this scenario.

The authors use MODIS EVI data in their analysis. Using remote sensing data would also allow them to quantify the vegetation response at other rewetted peatland sites in the region and would make their findings more generalizable. If near-natural fens exist in the region, the authors could also use a "reference site" to analyse the different vegetation responses between a rewetted and an intact system. I think this approach would provide additional evidence that the observed ecosystem responses can be generalised.

The authors quantify the immediate drought effects on vegetation dynamics and CO2 fluxes. However, as they point out in the discussion, it remains unclear what the long-term effects of this drought will be. Will the newly established vegetation survive in a following year with extended flooding? They discuss this issue briefly in their last paragraph. However, I think it would be beneficial to at least assess how EVI and/or CO2 fluxes in 2019 were affected by the 2018 drought if such an analysis is feasible.

Other comments Line 25: Consider using "short-term climate warming".

Line 50: Are there any drought studies for fens? It would be helpful to shortly summarise the current knowledge of drought impacts on carbon cycling in fens. Here are a few examples of fen studies:

Knorr et al. (2008): https://doi.org/10.1016/j.soilbio.2008.03.019

Robroek et al. (2017): https://royalsocietypublishing.org/doi/10.1098/rsos.170449

Olefeldt et al. (2017): https://doi.org/10.1111/gcb.13612

Schreader et al. (1998): https://doi.org/10.1029/98GB02738

Line 153: What was the overlap between eddy covariance flux footprint and MODIS pixel?

Line 163: Is it possible to quantify how much the increased light availability contributed to enhanced GPP? This could be done by comparing light-response curves. Even with similar light response curves, GPP could be different in 2018 due to differences in light availability.

Line 220: The authors could consider comparing so-called carbon uptake periods between the drought year and other years:

For example: Fu et al. (2017): https://doi.org/10.1016/j.agrformet.2017.05.009

Fig. 5c: Adding a 7-day moving average line to the graph would make the seasonal variations more visible.

---

## Author Comment (AC1) · 11 Sep 2020

General response: Dear editor, dear reviewers, your thoughtful comments and constructive suggestions will be extremely helpful in further improving the manuscript. In particular, the reviewer's suggestion to include CH4 data will broaden our perspective on drought effects in our study site (see new Figure 1). We will add a statement on the relevance of CH4 emissions for short-term climate effects due to rewetting in the introduction and add CH4 flux data from 2011 onwards to the study. As explained below, our data do not allow us to derive more information about possible effects during the post-drought year 2019, but we agree, that we could use the existing data more efficiently to extend our mechanistic understanding of drought-related processes. We will therefore test empirical modelling approaches that include (i) multiple regression using carbon uptake periods as a potential control variable and (ii) light use efficiency modelling. The reviewer's suggestions were also very helpful to stimulate new thoughts on the practical relevance of our study, which will be included in the Introduction and the Discussion section. As an example, we will relate our study to existing uncertainties in nature-based climate solutions to achieve the mitigation targets under a changing climate. Furthermore, our data suggests the importance of peatland rewetting to create hydrological retention areas as important prerequisite for landscapes resilient to climate change. Kind regards, Franziska Koebsch/Florian Beyer

In the following we respond to the individual comments of Reviewer 1 (RC1):

Comment 1 We agree: adding more context on nature-based climate solutions and the existing uncertainty to reach the mitigation targets under a changing climate would certainly increase the impact of the study. The reviewer's literature suggestions will be very helpful in amending the introduction accordingly.

Comment 2 Although the primary climate mitigation effect in peatland rewetting is due to the switch from a $CO_2$ source to a $CO_2$ sink, we decided to add $CH_4$ flux data to our study (see new Figure 1). Therefore, we will add to the introduction a description of the role of $CH_4$ in occasional droughts. We will also add a passage on $N_2O$, although we cannot provide $N_2O$ data from the drought period. We have measured $N_2O$ fluxes in 2009 and 2010, in the last year of drainage (dry conditions) and in the first year of rewetting (wet conditions). These data indicate that $N_2O$ fluxes were negligible under both hydrological regimes.

Comment 3 We understand that the addition of 2019 data in general would be helpful to better constrain carry-over effects of the drought. However, in January 2019 the area was flooded with brackish water from the adjacent Baltic Sea, which substantially affected the biogeochemistry of the peatland including vegetation and greenhouse gas

exchange. The conditions in 2019 will therefore be determined by both brackish water intrusion and possible effects after the drought, and we are unfortunately not able to clearly assign the observations in 2019 to either of these two factors. In order to prevent false conclusions on post-drought effects, we refrain from including 2019 data.

Comment 4 We agree that the observation data provide a good opportunity to deepen our mechanistic understanding on drought effects. We would therefore apply the following approaches and include the results in our manuscript: - a multiple regression model that allows the comparison of effect sizes as indications for varying sensitivities over different observation years -a light use efficiency model to better constrain potential drought-related limitations in plant photosynthesis

Comment 5 Line 27 We chose to include methane data (see new Figure 1), as another relevant greenhouse gas and present the relevance of methane for short-term climate impacts in the introduction.

Comment 6 Line 37 Indeed, our data suggest that there is a distinct reduction in methane emission after water levels decreased below surface. These data will be included in the study (see new Figure 1).
* * *
[Figure]

**Fig. 1.**

[Figure]

**Fig. 2.**

[Figure]

**Fig. 3.**

[Figure]

[Figure]

**Fig. 4.**

---

## Author Comment (AC2) · 11 Sep 2020

General response: Dear editor, dear reviewers, your thoughtful comments and constructive suggestions will be extremely helpful in further improving the manuscript. In particular, the reviewer's suggestion to include CH4 data will broaden our perspective on drought effects in our study site (see new Figure 1). We will add a statement on the relevance of CH4 emissions for short-term climate effects due to rewetting in the introduction and add CH4 flux data from 2011 onwards to the study. As explained below, our data do not allow us to derive more information about possible effects during the post-drought year 2019, but we agree, that we could use the existing data more effi-

ciently to extend our mechanistic understanding of drought-related processes. We will therefore test empirical modelling approaches that include (i) multiple regression using carbon uptake periods as a potential control variable and (ii) light use efficiency modelling. The reviewer's suggestions were also very helpful to stimulate new thoughts on the practical relevance of our study, which will be included in the Introduction and the Discussion section. As an example, we will relate our study to existing uncertainties in nature-based climate solutions to achieve the mitigation targets under a changing climate. Furthermore, our data suggests the importance of peatland rewetting to create hydrological retention areas as important prerequisite for landscapes resilient to climate change. Kind regards, Franziska Koebsch/Florian Beyer

In the following we respond to the individual comments of Reviewer 2 (RC2):

Comment 1 We think that the reviewer is right in his/her suggestion that the filled water reservoirs from last year's high rainfall contributed to the postponement of the hydrological drought and could thereby buffer the effect of the meteorological drought. The restoration of minerotrophic fens creates hydrological retention areas that slow down runoff, keep the water in the landscape longer and, additionally, decrease the surface temperature (Hemes et al. 2018). We think that this is another important plea for peatland rewetting that should be included in the manuscript: Not only showed the studied fen regulatory mechanisms to cope with temporary droughts, the restoration of fens also increases the resilience to drought on landscape level. Hemes, K. S., Eichelmann, E., Chamberlain, S. D., Knox, S. H., Oikawa, P. Y., Sturtevant, C., ... & Baldocchi, D. D. (2018). A unique combination of aerodynamic and surface properties contribute to surface cooling in restored wetlands of the Sacramento‐San Joaquin Delta, California. Journal of Geophysical Research: Biogeosciences, 123(7), 2072-2090.

Comment 2 You are absolutely right that remote sensing is a suitable tool to scale the described processes. However, we think that the major benefit of this study is the long-term reference data set on vegetation development and greenhouse gas exchange, which allows to discriminate the effects of the summer drought 2018 from climatenormal years. We think that, by including additional data on CH4 emissions (see new Figure 1) and empirical modelling, we can provide a deeper understanding on the drought-related processes at this study site.

Comment 3 (Same answer like Comment 3 of Reviewer 1) We understand that the addition of 2019 data in general would be helpful to better constrain carry-over effects of the drought. However, in January 2019 the area was flooded with brackish water from the adjacent Baltic Sea, which substantially affected the biogeochemistry of the peatland including vegetation and greenhouse gas exchange. The conditions in 2019 will therefore be determined by both brackish water intrusion and possible effects after the drought, and we are unfortunately not able to clearly assign the observations in 2019 to either of these two factors. In order to prevent false conclusions on post-drought effects, we refrain from including 2019 data.

Comment 4 Your comment is in line with reviewer 1 and as we will add CH4 flux data in this study we will provide a more detailed description of the climate impact of CH4 emissions.

Comment 5 Many thanks for your hint and the links. These studies do indeed contain some valuable information on drought processes in fens, which we will add to the introduction. Nevertheless, this selection also shows that most of our knowledge about drought effects in fens comes from treatment experiments and pristine sites. There are few natural observations, especially of restored fens, whose ecosystem functions and drought response mechanisms might substantially differ from those of natural sites.

Comment 6 You can see the overlap in Figure B2 (Modis Grid in blue, Eddy footprint in yellow). We'll add one sentence in line 157:"(. . .)Values were filtered according to pixel reliability and pixel-wise quality assessment and data gaps were subsequently filled by linear interpolation. The used MODIS pixel and the 90% footprint of the Eddy Tower are almost completely overlapping, as shown in Figure 2B.

Comment 7 We think this is a great idea to enhance our mechanistic understanding on

drought effects on GEP. In addition to multiple regression, light use efficiencies will be one modelling approach to be included in this study.

Comment 8 We agree that carbon uptake periods can provide additional insights into the processes that control the CH4 and CO2 exchange during drought. We will therefore implement carbon uptake periods as potential control variable in our multiple regression model.

Comment 9: All Figures of GEP, Reco, NEE and CH4 are now changed according to the suggestions of RC2 (7 days moving average of 2018 (black) as well as for the mean pre-drought period (dark gray)). Please see the new Figures.
* * *
[Figure]

**Fig. 1.**

[Figure]

**Fig. 2.**

[Figure]

**Fig. 3.**

[Figure]

**Fig. 4.**

---

## Author Response (AR1)

General response:

Dear editor, dear reviewers,
your thoughtful comments and constructive suggestions will be extremely helpful in further improving the manuscript. In particular, the reviewer's suggestion to include $CH_4$ data will broaden our perspective on drought effects in our study site. We
5  will add a statement on the relevance of $CH_4$ emissions for short-term climate effects due to rewetting in the introduction and add $CH_4$ flux data from 2011 onwards to the study. As explained below, our data do not allow us to derive more information about possible effects during the post-drought year 2019, but we agree, that we could use the existing data more efficiently to extend our mechanistic understanding of drought-related processes. We therefore added a light use efficiency model and carbon uptake periods as additional environmental forcing that help to better understand the biophysical mechanisms driving the $CO_2$
10  exchange under drought. Further, since our study is not suited to address post-drought effects, we provide some hypotheses on future development trajectories and their implications for the $CO_2$ and $CH_4$ exchange. The reviewer's suggestions were also very helpful to stimulate new thoughts on the practical relevance of our study, which will be included in the Introduction and the Discussion section. As an example, we will relate our study to existing uncertainties in nature-based climate solutions to achieve the mitigation targets under a changing climate. Furthermore, our data suggests the importance of peatland rewetting
15  to create hydrological retention areas as important prerequisite for landscapes resilient to climate change. Next, we provide our specific answers to the individual comments of the reviewers. This is followed by our edited manuscript, in which we have marked all changed text parts in red. We have also improved Figures 2, 3, 5 as well as B2 and added additional content.
Kind regards,
Franziska Koebsch/Florian Beyer

20  Authors comments to Reviewer 1:

Consider adding more context, in the intro, concerning why re-wetted peatlands could be an important climate change mitigation strategy. One important consideration is that many 'natural climate solution' potential portfolios often lack any consideration for how future climate change and disturbance regimes will impact the potential enhanced (or avoided) sequestration. There is an opportunity to better make the case of how crucial it is to use natural experiments like this to understand
25  the implications of disturbance on C sink potential of restored landscapes. Some references on NCS and peatlands:

*Reply: We strengthened the relevance of our study by presenting the topic in the light of nature climate solutions and the inherent uncertainty to reach the mitigation goals under a changing climate. This is also well in line with a recent study emphasizing the necessity of peatland rewetting to re-establish the net $CO_2$ sink function of the terrestrial land system (Humpenöder et al., 2020). The new content is presented in the introduction (line 166f.) and the conclusion chapter (line 487f. and line 523).*

30  Literature
Humpenöder, F., Karstens, K., Lotze-Campen, H., Leifeld, J., Menichetti, L., Barthelmes, A., and Popp, A.: Peatland protection and restoration are key for climate change mitigation, Environmental Research Letters, 15, 104 093, https://doi.org/-10.1088/1748-9326/abae2a, 2020.
Runkle, B. R. K., Suvocarev, K., Reba, M. L., Reavis, C. W., Smith, S. F., Chiu, Y.-L., and Fong, B.: Methane Emission
35  Reductions fromthe Alternate Wetting and Drying of Rice Fields Detected Using the Eddy Covariance Method, Environmental Science & Technology, 53,671–681, https://doi.org/10.1021/acs.est.8b05535, 2019.

Get more explicit about methane and $N_2O$, two other important GHG's that are quite dynamic in wetland/peatland systems, especially during drawdowns:

*Reply: Although the primary climate mitigation effect in peatland rewetting is due to the switch from a $CO_2$ source to a $CO_2$*
40  *sink, we decided to add $CH_4$ flux data to our study. Therefore, we added to the introduction a description of the role of $CH_4$ in occasional droughts (line 185f.). Further, we present and discuss the $CH_4$ flux patterns, as they occurred during the drought (line 431f.) and elucidate potential implications for the climate mitigation prospects of peatland rewetting (line 485f.). In this*

*regard, we found the reference suggestion of Runkle et al. (2019) very helpful to link our findings on temporary drought to the wet-dry-cycles deliberately introduced in rice cultivation to reduce $CH_4$ emissions (line 486). We also added a passage on $N_2O$ in the results & discussion part (line 443f.), although we cannot provide $N_2O$ data from the drought period. We have measured $N_2O$ fluxes in 2009 and 2010, in the last year of drainage (dry conditions) and in the first year of rewetting (wet conditions). These data indicate that $N_2O$ fluxes were negligible under both hydrological regimes.*

3. More discussion of future legacy effects.

*Reply: We understand that the addition of 2019 data in general would be helpful to better constrain carry-over effects of the drought. However, in January 2019 the area was flooded with brackish water from the adjacent Baltic Sea, which substantially affected the biogeochemistry of the peatland including vegetation and greenhouse gas exchange. The conditions in 2019 will therefore be determined by both brackish water intrusion and possible effects after the drought, and we are unfortunately not able to clearly assign the observations in 2019 to either of these two factors. In order to prevent false conclusions on post-drought effects, we refrain from including 2019 data.*

4. Better diagnosis of the mechanistic biophysical drivers of the biogeochemical changes. It is unclear what the specific biophysical cause of the reduced GEP and enhanced $R_{eco}$ is during the drought ...:

*Reply: We agree that the observation data provide a good opportunity to deepen our mechanistic understanding on drought effects. We therefore added fPAR, the fraction of absorbed photosynthetically active radiation as additional vegetation index and a light use efficiency (LUE) model to better constrain potential drought-related limitations in plant photosynthesis. The LUE model indicated stomata closure as potential biophysical mechanism for the reduction of GEP in the first phase of the drought period (line 420f.). For the 2nd phase of the drought period, LUE stressed the efficiency of $CO_2$ assimilation for the vigorous biomass production of pioneer species and the reinforcing effect of high temperatures to enhance the capacity of photosynthetic $CO_2$ assimilation late in the season (line 410f.)*

Line 27: "Therefore, climate mitigation measures in peatlands need to focus primarily on the reduction of the $CO_2$ source".) + Line 37: "drought implies a lowering of the ground water level" What about impacts on $CH_4$ evolution if redox conditions change?

*Reply: We chose to include methane data, as another relevant greenhouse gas and present the relevance of methane for short-term climate impacts in the introduction (line 185f.).*

Authors comments to Reviewer 2:

I am wondering if the results of this study can be generalised. The authors present an interesting case study, but it could be that specific site characteristics and the specific drought characteristics were mainly responsible for the observed responses in $CO_2$ fluxes. The authors mention that the water table in spring 2018 was unusually high due to the previous year's high precipitation. This apparently led to a discrepancy between meteorological drought (mainly in May) and hydrological drought (from August on when the water table dropped below the surface). This specific setting might be responsible for the high GPP rates early in the growing season (see Fig. 5a). If the meteorological and hydrological drought would co-occur, a negative effect on GPP could be possible potentially leading to the fen becoming a net $CO_2$ source. This hypothesis might be difficult to test with the existing data, but the authors might consider discussing this scenario.

*Reply: We think that the reviewer is right in his/her suggestion that the filled water reservoirs from last year's high rainfall contributed to the postponement of the hydrological drought and could thereby buffer the effect of the meteorological drought. This is now discussed in line 477. Nevertheless, we think that the inherent hydrological sink function is a common feature for fens, and actually strengthens the representativity of our study. Therefore, we point out the necessity to restore the natural hydrological sink function to create peatlands resilient to drought events (line 482f.). Further, we characterize the study as*

*important starting point for further research and describe the circumstances under which our results could be transferred to other cases (line 519f.)*

85    The authors use MODIS EVI data in their analysis. Using remote sensing data wouldalso allow them to quantify the vegetation response at other rewetted peatland sitesin the region and would make their findings more generalizable. If near-natural fensexist in the region, the authors could also use a "reference site" to analyse the differentvegetation responses between a rewetted and an intact system. I think this approachwould provide additional evidence that the observed ecosystem responses can be generalised

90    *Reply: You are absolutely right that remote sensing is a suitable tool to scale the described processes. However, we think that the major benefit of this study is the long-term reference data set on vegetation development and greenhouse gas exchange, which allows to discriminate the effects of the summer drought 2018 from climate-normal years. We think that, by including additional data on $CH_4$ emissions, light use efficiency modelling and carbon uptake periods as additional environmental forcing, we can provide a deeper understanding on the drought-related processes at this study site.*

95    The authors quantify the immediate drought effects on vegetation dynamics and $CO_2$ fluxes. However, as they point out in the discussion, it remains unclear what the long-term effects of this drought will be. Will the newly established vegetation survive ina following year with extended flooding? They discuss this issue briefly in their lastparagraph. However, I think it would be beneficial to at least assess how EVI and/or $CO_2$ fluxes in 2019 were affected by the 2018 drought if such an analysis is feasible

100    *Reply: We understand the demand to add data from the following year(s). However, as described in response to reviewer 1, we refrain from including 2019 data, because the site had been flooded with brackish water in January 2019 and we want to prevent false conclusions on post-drought effects. Instead, we added some considerations on possible future scenarios and their implications for the $CO_2$ and $CH_4$ exchange (line 496f.)*

    Other comments Line 25: Consider using "short-term climate warming"

105    *Reply: We have reorganized the entire part to desribe the warming effect of $CH_4$ emissions in more detail.*

    Line 50: Are there any drought studies for fens? It would be helpful to shortly summarise the current knowledge of drought impacts on carbon cycling in fens. Here area few examples of fen studies:
Knorr et al. (2008): https://doi.org/10.1016/j.soilbio.2008.03.019
Robroek et al. (2017): https://royalsocietypublishing.org/doi/10.1098/rsos.170449O
110   Leifeldt et al. (2017): https://doi.org/10.1111/gcb.13612
Schreader et al. (1998): https://doi.org/10.1029/98GB02738

    *Reply: Thanks for the literature suggestions, these are now complementing the introduction and add to a more comprehensive description of drought effects in both, bogs and fen (line 181f.)*

    Line 153: What was the overlap between eddy covariance flux footprint and MODISpixel?

115    *Reply: We computed the footprint climatology and present the overlap with the MODIS products in Figure B2. According to the resulting footprint climatology, 90 % of the measured gas exchange comes from within 200 m distance around the eddy covariance tower which is well within the MODIS grid cell.*

Line 163: Is it possible to quantify how much the increased light availability contributedto enhanced GPP? This could be done by comparing light-response curves. Even withsimilar light response curves, GPP could be different in 2018 due to differences in lightavailability.

*Reply: We used a simple light use efficiency (LUE) model to provide more insights into the biophysical mechanisms controlling GEP during drought. This model is also indicative for the capacity of photosynthetic $CO_2$ assimilation the maximum photosynthesis rate at light saturation, both of which could exlain the high GEP rates in autumn 2018 (line 425f.)*

Line 220: The authors could consider comparing so-called carbon uptake periods be-tween the drought year and other years: For example: Fu et al. (2017): https://doi.org/10.1016/j.agrformet.2017.05.009

*Reply: We thank for this valueable hint. We estimated the start, end and length of the CUP and it turned out that the high GEP rates in late autumn 2018 coincided will an extension of the CUP by 26 days in comparison to average years.We added this fact into the discussion (line 427f.)*

Fig. 5c: Adding a 7-day moving average line to the graph would make the seasonalvariations more visible.

*Reply: We added auch a moving average line to our figures and agree that it substantially improves the readability of the figures*

[revised manuscript text omitted]

---

## Author Response (AR2)

General response:

Dear editor, dear referees,

Thank you very much for your valuable comments. Once again, they were very helpful in improving our manuscript. We have thoughtfully reviewed all comments and incorporated them where appropriate. We have also corrected some minor spelling
5   mistakes. All changes made are marked in red in the following document.

After careful consideration, we decided not to include the Chamberlain et al. (2019) study in our manuscript because the focus of Chamberlain et al. (2019) is on a salinization effect due to drought. In contrast, salinities measured at our study site during drought did not exceed 3 $ppt$. Therefore, we think that the vegetation and greenhouse gas dynamics were related to the processes described in the manuscript rather than to a salinization effect.

10   Kind regards,

Franziska Koebsch/Florian Beyer

[revised manuscript text omitted]